# Corollary discharge in precerebellar nuclei of sleeping infant rats

Didhiti Mukherjee[1,2], Greta Sokoloff[1,2,3], Mark S Blumberg[1,2,3,4,5]*

[1]Department of Psychological and Brain Sciences, University of Iowa, Iowa, United States; [2]Delta Center, University of Iowa, Iowa, United States; [3]Iowa Neuroscience Institute, University of Iowa, Iowa, United States; [4]Interdisciplinary Graduate Program in Neuroscience, University of Iowa, Iowa, United States; [5]Department of Biology, University of Iowa, Iowa, United States

**Abstract** In week-old rats, somatosensory input arises predominantly from external stimuli or from sensory feedback (reafference) associated with myoclonic twitches during active sleep. A previous study suggested that the brainstem motor structures that produce twitches also send motor copies (or corollary discharge, CD) to the cerebellum. We tested this possibility by recording from two precerebellar nuclei—the inferior olive (IO) and lateral reticular nucleus (LRN). In most IO and LRN neurons, twitch-related activity peaked sharply around twitch onset, consistent with CD. Next, we identified twitch-production areas in the midbrain that project independently to the IO and LRN. Finally, we blocked calcium-activated slow potassium (SK) channels in the IO to explain how broadly tuned brainstem motor signals can be transformed into precise CD signals. We conclude that the precerebellar nuclei convey a diversity of sleep-related neural activity to the developing cerebellum to enable processing of convergent input from CD and reafferent signals.
DOI: https://doi.org/10.7554/eLife.38213.001

*For correspondence:
mark-blumberg@uiowa.edu

**Competing interests:** The authors declare that no competing interests exist.

## Introduction

The sensorimotor systems of diverse vertebrate and invertebrate species distinguish signals arising from self-generated movements (i.e., reafference) from those arising from other-generated movements (i.e., exafference; *Cullen, 2004*). To make this distinction, motor structures generate copies of motor commands, referred to as corollary discharge (CD; *Crapse and Sommer, 2008*; *Poulet and Hedwig, 2007*). CD is conveyed to non-motor structures to inform them of the imminent arrival of reafference arising from self-generated movements (*Crapse and Sommer, 2008*). By comparing the two signals, animals are able to distinguish between self-produced and other-produced movements.

Self-produced movements are not restricted to periods of wakefulness, especially during development. Infants produce brief, discrete, jerky movements of skeletal muscles during active sleep (AS or REM sleep), a predominant behavioral state during early infancy (*Jouvet-Mounier et al., 1969*; *Roffwarg et al., 1966*). These spontaneous movements, called myoclonic twitches, are most abundant and conspicuous in developing mammals (*Blumberg et al., 2013*; *Gramsbergen et al., 1970*; *Jouvet-Mounier et al., 1969*; *Roffwarg et al., 1966*).

Revealing the similarities and differences between twitches and wake movements is important for understanding the contribution that each type of movement makes to the development of the sensorimotor system. For example, in addition to their very different kinematic properties, twitches and wake movements differ in how they are processed by the infant brain. Specifically, in week-old rats the external cuneate nucleus (ECN), a medullary nucleus that receives proprioceptive input from the forelimbs, actively inhibits reafference arising from wake-related limb movements but not those arising from twitches (*Tiriac and Blumberg, 2016*). This state-dependent gating by the ECN suggested the selective engagement of a CD mechanism during wake movements and its suspension during

twitching. It remained unclear, however, whether the suspension of the gating mechanism reflected the absence of a twitch-related CD signal or the inhibition of CD's effects within the ECN. Resolving the question of whether CD accompanies twitching would be an important step toward understanding the functional significance of CD in early development. Indeed, if twitches contribute to the process whereby limbs are assimilated into the emerging body schema (*Blumberg and Dooley, 2017*), then one would expect CD—which underlies the capacity to distinguish self from other—to be critical to that process.

There is some indirect evidence that twitches are accompanied by CD. In week-old rats, twitches trigger both complex spikes (arising from climbing fibers) and simple spikes (arising from mossy fibers) in cerebellar Purkinje cells (*Sokoloff et al., 2015a*). These neural events were detected at latencies that were likely too short to be reafferent signals arising from the periphery (*Puro and Woodward, 1977a*; *Puro and Woodward, 1977b*). Accordingly, it is possible that the motor structures that produce twitches also convey CD to the cerebellum, as occurs with waking movements in adults (*Azim and Alstermark, 2015*; *Azim et al., 2014*).

If a twitch-related CD signal reaches the cerebellar cortex, it must be conveyed through the pre-cerebellar nuclei. The inferior olive (IO) is a good candidate structure for such a CD signal. First, it is the sole source of climbing fibers to cerebellar cortex and is therefore responsible for the triggering of complex spikes (*Ruigrok et al., 2014*). Second, midbrain motor structures project directly to the IO (*De Zeeuw et al., 1998*). Finally, the IO fires precisely at the onset of self-generated movements in waking adults (*Keating and Thach, 1995*; *Welsh et al., 1995*).

With respect to mossy fibers, there are a few major candidate structures to consider (*Ruigrok et al., 2014*). First, the pontine nucleus is an unlikely source of CD in week-old rats because it receives descending input from motor cortex (*Lee and Mihailoff, 1990*), which does not contribute to the production of twitches (*Blumberg, 2010*; *Kreider and Blumberg, 2000*). Second, the ECN can also be ruled out as a source of CD because it processes twitch-related reafference exclusively (*Tiriac and Blumberg, 2016*). Finally, the lateral reticular nucleus (LRN) is a possible candidate because it receives both sensory input from the limbs and motor input from midbrain structures, including the red nucleus (*Alstermark and Ekerot, 2013*; *Pivetta et al., 2014*). Moreover, the LRN has been implicated in processing CD associated with self-generated movements in adults (*Alstermark and Ekerot, 2015*; *Arshavsky et al., 1978*).

Accordingly, we recorded neural activity in the IO and LRN in postnatal day (P) 7–9 (hereafter P8) rats. Relying on three proposed criteria for identifying CD signals (*Poulet and Hedwig, 2007*; *Sommer and Wurtz, 2008*), we first show that neurons within the IO and LRN sharply increase their activity within ±10 ms of twitch onset; such activity is not easily attributable to either motor or reafferent activity. Second, we show that the twitch-related IO and LRN activity originates in midbrain structures that contribute to motor outflow. Because neither the IO nor LRN plays a direct role in the production of movement (*Gellman et al., 1985*; *Ruigrok et al., 2014*), a third CD criterion is satisfied. Finally, we test the hypothesis that calcium-activated slow potassium (SK) channels are responsible for converting the broadly tuned motor signal arising from the midbrain into a sharply tuned CD signal. To our knowledge, these findings provide the first direct neurophysiological evidence of a CD signal in an infant mammal.

## Results

### IO activity predominates during active sleep

We recorded IO activity in unanesthetized head-fixed pups as they cycled spontaneously between sleep and wake with their limbs dangling freely (n = 20 pups, 37 units, 1–4 units/pup). Electromyography (EMG) and behavioral scoring were used to identify behavioral state and detect sleep and wake movements (*Blumberg et al., 2015*; *Figure 1A*).

Electrode placement within the IO was confirmed histologically (*Figure 1B*). Recording sites were located within the dorsal accessory olive (DAO; n = 19 units across 12 pups) or the medial accessory olive (MAO) and the principal olive (PO; n = 18 units across eight pups; *Figure 1C*). Overall, unit activity was phasic and largely restricted to periods of AS; unit activity often decreased immediately after the onset of active wake (AW). Sparse activity was observed during behavioral quiescence (BQ), which is a period of low muscle tone interposed between AW and AS (*Figure 1D*). The

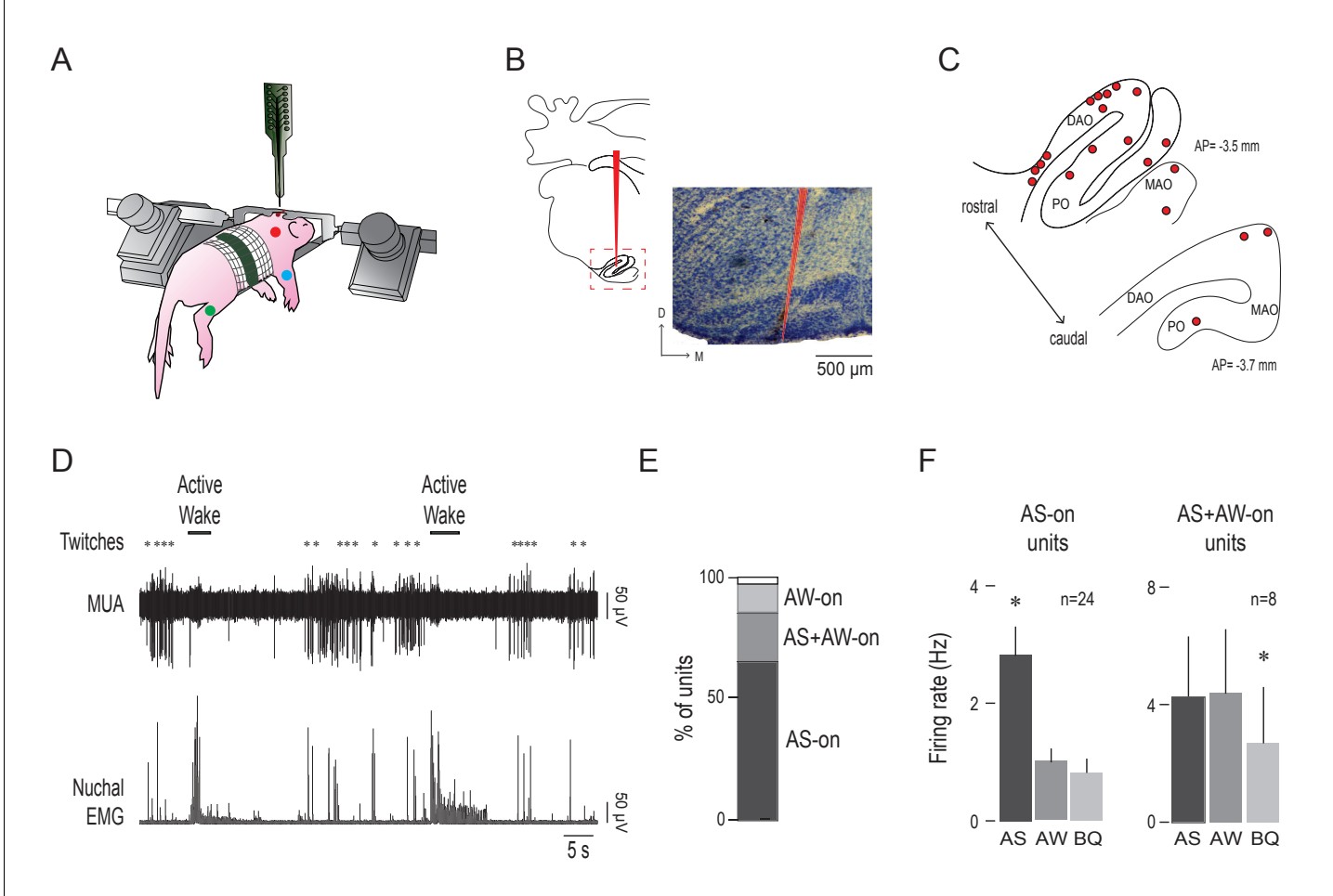

**Figure 1.** Olivary activity predominates during active sleep. (**A**) Illustration of a head-fixed rat pup in a recording apparatus instrumented with nuchal (red), forelimb (blue), and hindlimb (green) EMG electrodes. (**B**) Left: Reconstruction of a representative electrode placement within the IO (red line). Red dashed box circumscribes the IO. Right: Representative coronal Nissl-stained brain section. Red line is the trace of a DiI-coated electrode placed within the IO. D: dorsal; M: medial (**C**) Electrode placements (red circles) within the IO in two coronal sections across all subjects. DAO: dorsal accessory olive; MAO: medial accessory olive; PO: principal olive; AP: antero-posterior distance in relation to lambda. (**D**) Representative recording of rectified nuchal EMG activity and multiunit activity (MUA) in the IO during spontaneous sleep-wake cycling. Asterisks denote twitches and horizontal bars denote periods of active wake movements as scored by the experimenter. (**E**) Stacked plot showing the percentage of IO units that were AS-on, AS+AW-on, and AW-on. (**F**) Mean (+SEM) firing rates of AS-on (left) and AS+AW-on (right) units across behavioral states. Each individual unit included in these means was significantly state dependent. * significant difference from the other two behavioral states, p < 0.008. AS: active sleep; AW: active wake; BQ: behavioral quiescence.

DOI: https://doi.org/10.7554/eLife.38213.002

The following source data is available for figure 1:

**Source data 1.** Source data for panels E and F.

DOI: https://doi.org/10.7554/eLife.38213.003

majority of IO units were AS-on (23/35; *Figure 1E*) and the mean firing rate of the AS-on units (2.8 ± 0.5 Hz) was approximately three times higher during AS than during the other two states (p < 0.0001; *Figure 1F*, left). A smaller proportion of units was AS+AW-on (9/35); the mean firing rates during AS and AW (4.3 ± 2.1 Hz and 4.4 ± 2.2 Hz, respectively) were approximately two times higher than that during BQ (p < 0.004; *Figure 1F*, right). Only 2/35 units were AW-on. Two IO units were excluded from state analysis due to movement artifact.

## IO neurons exhibit sharp activity peaks at twitch onset

The phasic IO activity clustered around periods of myoclonic twitching; therefore, we examined the temporal relationship between twitches and unit activity by creating perievent histograms (5-ms bins, 1-s windows) with unit activity triggered on twitch onset.

Previous studies have revealed two distinct patterns of twitch-triggered perievent histograms in sensorimotor structures. First, in a motor structure like the RN, unit activity increases 20–40 ms before the onset of a twitch (*Del Rio-Bermudez et al., 2015*). Second, in a sensory structure like the ECN, unit activity increases at least 10–50 ms after the onset of a twitch (*Tiriac and Blumberg, 2016*). In the IO, however, the majority of units (23/37) were active within ±10 ms of twitch onset (*Figure 2A–B*). This IO activity profile is strikingly different from that observed in motor and sensory structures from which we have previously recorded (*Figure 2C*). Also, the IO units that exhibited this profile were responsive primarily to nuchal and/or forelimb twitches and rarely to hindlimb twitches (*Figure 2—figure supplement 1A–D*). Finally, the characteristics of the neural responses recorded in the IO did not appear to differ across anatomical subdivisions.

There are three possible explanations for these sharply peaked activation patterns observed in IO units: (a) the IO is part of the motor pathway, (b) the IO receives reafference from twitches, and (c) the IO receives CD from a motor structure that produces twitches. With respect to (a), the IO, despite being implicated in the precise timing of motor behaviors (*De Zeeuw et al., 1998*), is not directly involved in the generation of movements (*Horn et al., 2004*; *Lang et al., 2017*). Although it receives afferent projections from motor areas (*Saint-Cyr, 1983*; *Saint-Cyr and Courville, 1981*), there are no efferent projections from the IO to spinal motor neurons. In fact, cerebellar climbing fibers comprise the sole efferent projection from the IO (*Ruigrok et al., 2014*). Consequently, in adults, stimulation of the IO does not evoke or modulate movements (*Gellman et al., 1985*).

With respect to (b), although the IO can receive short-latency reafferent signals (*Gellman et al., 1983*; *Sedgwick and Williams, 1967*), it is unlikely that reafference can account for the short-latency peaks observed here. Consider that for the structures in which we have seen clear evidence of twitch-related reafference (e.g. ECN), we have also seen clear exafferent responses (*Tiriac et al., 2014*; *Tiriac and Blumberg, 2016*). In contrast, in IO units that exhibited sharp peaks with a latency of ±10 ms, exafferent stimulation did not evoke significant increases in firing rate (see *Figure 2—figure supplement 1E*). Moreover, only a small number of IO units (7/37) exhibited twitch-triggered responses at latencies consistent with reafferent processing (>10 ms; *Figure 2—figure supplement 1F*). Thus, the signature feature of the majority of IO activity—a sharp peak centered on twitch onset—is consistent with the notion that the IO receives CD associated with the production of a twitch.

## LRN neurons exhibit two kinds of twitch-related activity

Based on research in adults (*Alstermark and Ekerot, 2015*; *Arshavsky et al., 1978*), we predicted that the LRN, like the IO, would exhibit CD-related activity. Moreover, because the LRN also receives sensory inputs from the limbs (*Figure 3A*), we expected to see evidence of reafference in that structure. To test these two possibilities, we next recorded spontaneous LRN activity in P8 rats across sleep and wake.

We confirmed electrode placements in the LRN (n = 27 units across 9 pups, 1–6 units/pup; *Figure 3B*). Similar to the IO, the unit activity in the LRN was phasic and restricted to periods of AS, particularly around twitches. LRN activity was sparse during BQ and often decreased immediately after AW onset (*Figure 3C*). The majority of LRN units (17/27) were AS-on (*Figure 3D*) and the mean firing rate of the AS-on units (1.4 ± 0.2 Hz) was approximately three times higher during AS than during AW or BQ (p =< 0.0005; *Figure 3E*, left). In addition, a smaller proportion of units (8/27 were AS +AW-on, and the mean firing rates during AS and AW (1.2 ± 0.26 Hz and 1.1 ± 0.26 Hz, respectively) were approximately three times higher than that during BQ (p < 0.02; *Figure 3E*, right). Only 2/27 units were AW-on.

Next, we assessed the temporal relationship between LRN unit activity and twitches by creating perievent histograms (5-ms bins, 1-s windows). Regardless of state dependency, the majority of LRN units (24/27) showed significant increases in firing rate in response to a twitch (*Figure 3F*). As predicted, we observed two different neural populations that exhibited distinct patterns of twitch-triggered activity. First, we found a subpopulation of LRN units (12/27) that, like the majority of IO units,

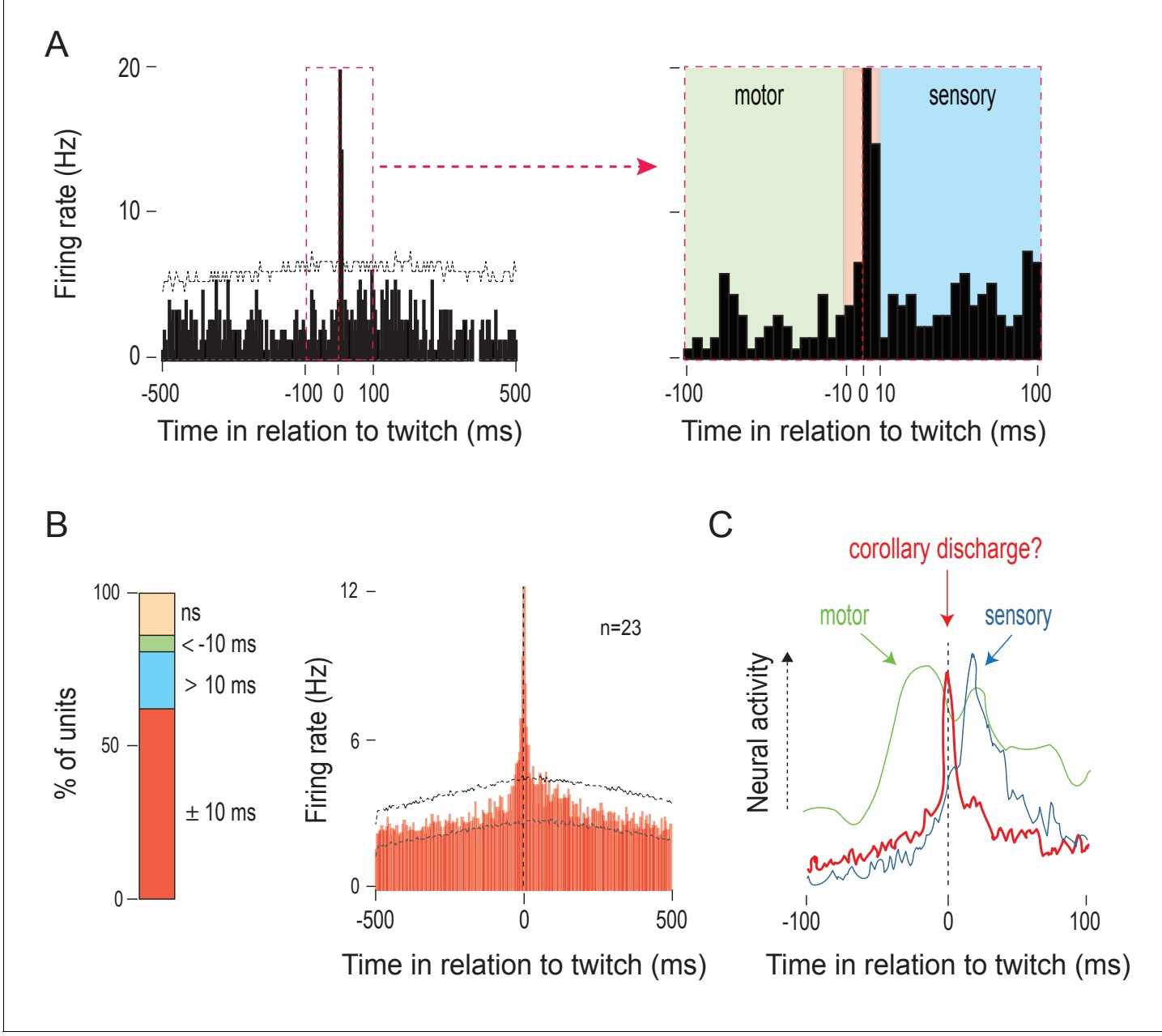

**Figure 2.** Twitches trigger sharp, short-latency olivary activity. (**A**) Left: Perievent histogram (5-ms bins) showing sharp, short-latency activity of a representative IO unit in relation to nuchal muscle twitches. Upper confidence band (p < 0.01 for each band) is indicated by the horizontal dashed line (lower confidence band is at zero). The red dashed box demarcates the ±100-ms time window around twitches. Right: IO unit activity within the ±100-ms period around twitches. Three time windows are shown: <-10 ms (green), ±10 ms (red), and >10 ms (blue), respectively. (**B**) Left: Stacked plot showing the percentage of IO units that exhibited significant increases in firing within the three time windows around twitches. Right: Perievent histogram (5-ms bins) showing IO unit activity in relation to twitches for those units that were significantly active within the ±10 ms time window. Data are pooled across 23 units and triggered on 6602 twitches. Upper and lower confidence bands (p < 0.01 for each band) are indicated by horizontal dashed lines. ns: not significant. (**C**) Illustrative comparison of IO activity in relation to twitches (red line; from B) with the neural activity of a representative motor structure (green line; data for the RN from *Del Rio-Bermudez et al., 2015*) and sensory structure (blue line; data for the ECN from *Tiriac and Blumberg, 2016*).

DOI: https://doi.org/10.7554/eLife.38213.004

The following source data and figure supplements are available for figure 2:

**Source data 1.** Source data for panels A-C.

DOI: https://doi.org/10.7554/eLife.38213.007

**Figure supplement 1.** IO units respond predominantly to nuchal and forelimb twitches.

*Figure 2 continued on next page*

*Figure 2 continued*

DOI: https://doi.org/10.7554/eLife.38213.005

**Figure supplement 1—source data 1.** Source data for panels A-F.

DOI: https://doi.org/10.7554/eLife.38213.006

exhibited a sharp peak within ±10 ms of twitch onset (*Figure 3G*, left); none of these LRN units responded to exafferent stimulation of the limbs (data not shown). Second, the remaining LRN units (12/27) exhibited broader twitch-related activity profiles consisting of a peak in activity around twitch onset (±10 ms) and/or a peak with a latency of >10 ms (*Figure 3G*, right). The latter peak is what is expected from a short-latency reafferent responses (*Tiriac and Blumberg, 2016*; *Tiriac et al., 2014*). Moreover, in 6 of these 12 units, exafferent stimulation of the limbs evoked increased firing rates with an average latency of 40 ms (*Figure 3H*).

## Non-overlapping regions in the mesodiencephalic junction (MDJ) project to the IO and LRN

The MDJ includes diverse structures, like the RN, that innervate spinal motor neurons (*De Zeeuw et al., 1998*; *Saint-Cyr and Courville, 1981*; *Lakke, 1997*; *Onodera and Hicks, 2009*; *Zuk et al., 1983*) and are therefore directly involved in the generation of movements (*Fukushima, 1991*; *Morris et al., 2015*; *Onodera and Hicks, 1996*; *Williams et al., 2014*). To determine whether MDJ neurons also project to the IO and LRN at P8, we performed retrograde tracing from each structure.

Wheat germ agglutinin (WGA) conjugated with Alexa Fluor 488 or 555 was microinjected into the IO or LRN. Retrograde tracing from the IO (n = 5; *Figure 4A*) revealed robust labeling of cell bodies in the MDJ, including diffuse areas around the fasciculus retroflexus (fr). Consistent with observations in adult rats, little or no labeling was observed in the RN or nucleus of Darkschewitsch (Dk, see *Ruigrok et al., 2014*). In contrast, retrograde tracing from the LRN (n = 4; *Figure 4B*) revealed robust labeling in the contralateral RN but not elsewhere in the MDJ. Moreover, in two of these four pups with LRN injections, a second tracer was also injected into the IO; in both of these pups with dual tracing, we again found that LRN-projecting cell bodies were located within the RN and also that IO-projecting cell bodies were located adjacent to the RN (*Figure 4C*). Altogether, these findings show that the IO and LRN receive projections from non-overlapping MDJ regions. Importantly, these results are consistent with those reported previously in adult rats (*Ruigrok et al., 2014*).

## MDJ stimulation causes limb movements and c-Fos activation in the IO and LRN

To assess functional connectivity between the MDJ and the IO or LRN, we electrically stimulated the RN (n = 4) and non-RN MDJ nuclei (n = 4) while monitoring forelimb and hindlimb movements in urethanized (1.5 mg/g, IP) P8 rats (*Figure 4—figure supplement 1A*). Subsequently, we performed immunohistochemistry to determine the expression of the c-Fos protein, a marker of neural activity (*Chung, 2015*), in the IO and LRN.

Stimulation of several non-RN MDJ nuclei evoked non-specific movements of the ipsilateral and contralateral forelimbs and hindlimbs and also resulted in c-Fos expression primarily in the ipsilateral IO (*Figure 4—figure supplement 1B*). In contrast, stimulation of the RN produced only discrete contralateral forelimb movements and resulted in c-Fos expression within and adjacent to the contralateral LRN, but not the IO (*Figure 4—figure supplement 1C*). These results indicate that MDJ nuclei are functionally connected to the IO and LRN at these ages.

It is possible that c-Fos activation in the IO and LRN was due to sensory feedback arising from the stimulated movements. However, we observed little or no c-Fos expression in sensory areas like the cuneate nucleus and ECN (data not shown).

## MDJ neurons adjacent to the RN are active before and after the production of twitches

It is known that the RN is involved in the production of twitches at P8 (*Del Rio-Bermudez et al., 2015*) and, as shown here, in the conveyance of CD to the LRN. Similarly, if MDJ neurons outside of the RN convey twitch-related CD to the IO, we would also expect these neurons to be involved in

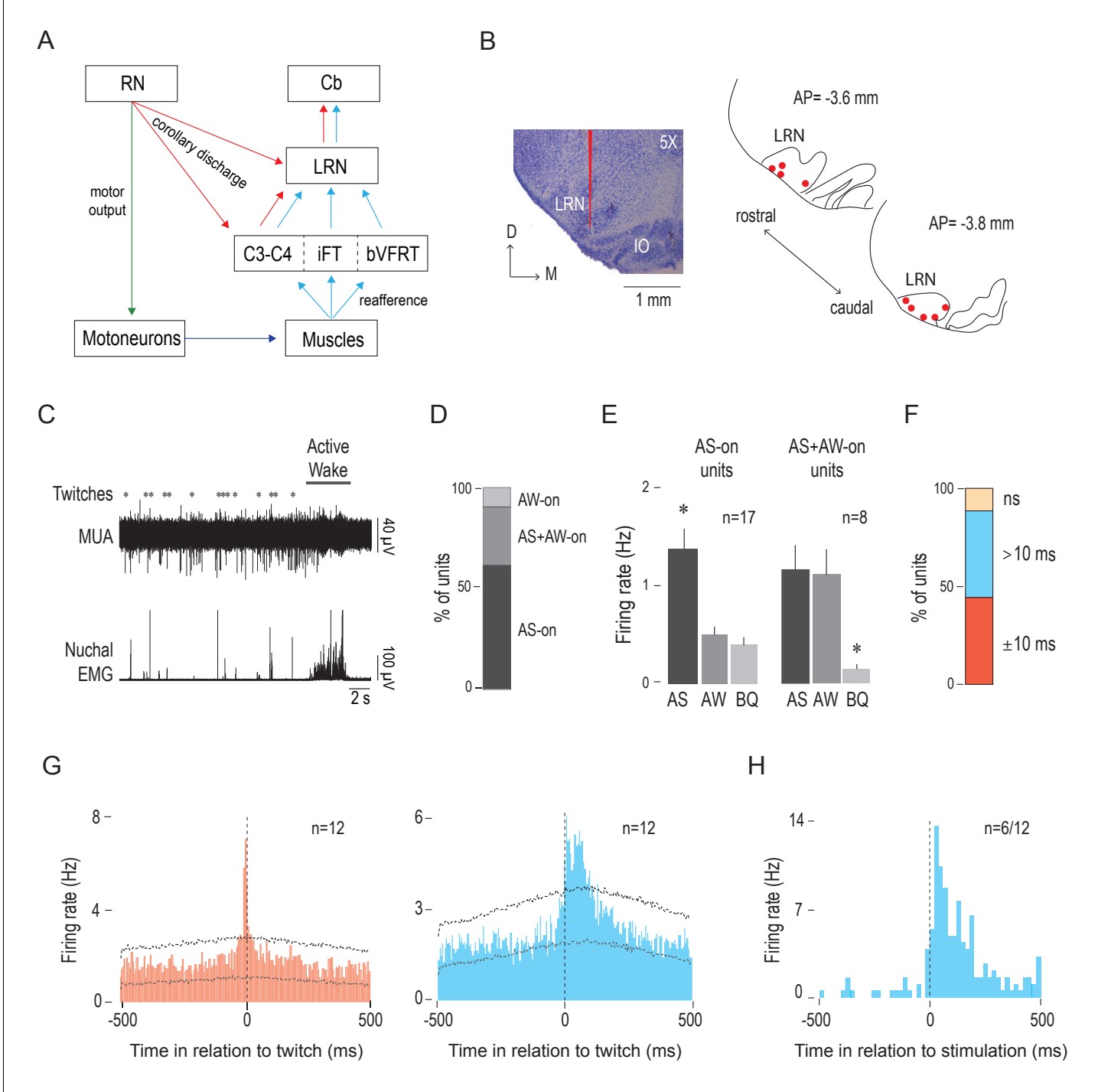

**Figure 3.** The LRN receives twitch-related corollary discharge and reafference signals. (A) Diagram depicting afferent and efferent connections of the LRN. Pathways conveying motor commands (green), reafference (blue), and corollary discharge (red) are shown (see *Alstermark and Ekerot, 2013*). C: cervical segment; Cb: cerebellum; iFT: ipsilateral forelimb tract; bVFRT: bilateral ventral flexor reflex tract. (B) Left: Representative coronal Nissl-stained brain section to show the trace of a DiI-coated electrode placed within the LRN (red line). Right: Electrode placements (red circles) within the LRN in two coronal sections across all P8 subjects (n = 9). D: dorsal; M: medial; AP: antero-posterior distance in relation to lambda. (C) Representative recording of rectified nuchal EMG activity and multiunit activity (MUA) in the LRN during spontaneous sleep-wake cycling. Asterisks denote twitches and the horizontal bar denotes a period of active wake movements as scored by the experimenter. (D) Stacked plot showing the percentage of LRN units that were AS-on, AS+AW-on, and AW-on. (E) Mean (+SEM) firing rates of AS-on (left) and AS+AW-on (right) units across behavioral states. Each individual unit included in these means was significantly state dependent. * significant difference from the other two behavioral states, p < 0.02. (F) Stacked plot showing the percentage of LRN units that significantly increased their firing rates within two time windows in relation to twitch onset: ±10

*Figure 3 continued on next page*

*Figure 3 continued*
ms (red) and >10 ms (blue). ns: not significant. (**G**) Perievent histograms (5-ms bins) showing LRN unit activity in relation to twitches. Left: Data pooled across 12 units (triggered on 3688 twitches) that significantly increased their activity in the ±10-ms time window (red). Right: Data pooled across the 12 units (triggered on 5264 twitches) that exhibited a significant peak in the >10-ms time window (blue). Upper and lower confidence bands (p < 0.01 for each band) are indicated by horizontal dashed lines. (**H**) Perievent histogram (20-ms bins) showing LRN unit activity in response to forelimb or hindlimb stimulation for those units (6/12) that significantly increased their activity in the >10-ms time window (blue histogram in G). Black vertical dashed line denotes stimulation onset as determined using EMG activity.
DOI: https://doi.org/10.7554/eLife.38213.008
The following source data is available for figure 3:

**Source data 1.** Source data for panels D-H.
DOI: https://doi.org/10.7554/eLife.38213.009

the production of twitches (*Figure 5A*). Therefore, we characterized the spontaneous activity of non-RN MDJ neurons in P8 rats during sleep and wake. We aimed to record in regions implicated earlier as projecting to the IO and, upon stimulation, producing limb movements (see *Figure 4* and *Figure 4—figure supplement 1*).

Electrode placements in the MDJ outside the RN were confirmed (n = 7 pups, 17 units, 1–5 units/pup; *Figure 5B*). The spontaneous activity of neurons in this region appeared mostly around twitches and wake movements (*Figure 5C*). When twitch-triggered perievent histograms (10-ms bins, 1-s windows) were created, we found that nearly all of the recorded units (15/17) showed significant twitch-dependent activity (*Figure 5D*).

The temporal relationship between neural activity and twitches revealed two primary subpopulations of units (*Figure 5D*): There were units that significantly increased their firing rates before the onset of a twitch (twitch-preceding) and units that significantly increased their firing rates after the onset of a twitch (twitch-following). The majority of the twitch-preceding units (10/17 across 5 pups) showed increase firing rates 10–70 ms before twitch onset (*Figure 5E*, left); the majority of these units (7/10) also exhibited increased firing rates 10–50 ms after twitch onset, indicative of reafference. In contrast, the twitch-following units (5/17 across 3 pups) only showed increased firing rates 20–40 ms after twitch onset (*Figure 5E*, right), suggesting that they only receive twitch-related reafference. In terms of firing pattern and latency, these neurons behave similarly to those described previously in the RN (*Figure 5F*; *Del Rio-Bermudez et al., 2015*).

We also analyzed unit activity in relation to wake movements, which occur much less frequently than twitches at this age. Overall, focusing on those units that exhibited significant twitch-related activity, we found that wake-related activity was relatively weak. First, of the 10 twitch-preceding non-RN MDJ units, 4 significantly increased their activity before the onset of wake movements (*Figure 5—figure supplement 1A*). Second, of the 21 IO units that exhibited twitch-related CD activity, only 1 significantly increased its activity within ±10 ms of the onset of wake movements (*Figure 5—figure supplement 1B*; two units were excluded due to movement artifact); similarly, only 1 of the 11 LRN units that exhibited twitch-related CD activity also increased its activity around wake-movement onset (*Figure 5—figure supplement 1C*; one unit was excluded due to movement artifact). Finally, of the 12 LRN units that exhibited significant sensory responses to twitches, 5 exhibited clear and significant sensory responses after the onset of wake movements (*Figure 5—figure supplement 1D*).

## Calcium-activated slow-potassium (SK) channels contribute to the sharp peak in IO activity

Having identified motor structures that send CD to the IO and LRN, we next sought to determine how a motor command with a broad twitch-preceding peak (see *Figure 5F*) is transformed into a sharp, precise peak around a twitch (see *Figure 2B*). We focused on the IO to address this question because of the reliably high percentage of units that exhibit twitch-related CD.

In the adult IO, SK channels prevent temporal summation of excitatory presynaptic inputs (*Garden et al., 2017*). SK channels are also expressed early in development (*Gymnopoulos et al., 2014*). Because afferent projections from the MDJ to the IO are excitatory, we hypothesized that twitch-related CD conveyed to the IO is accompanied by the opening of SK channels, thereby

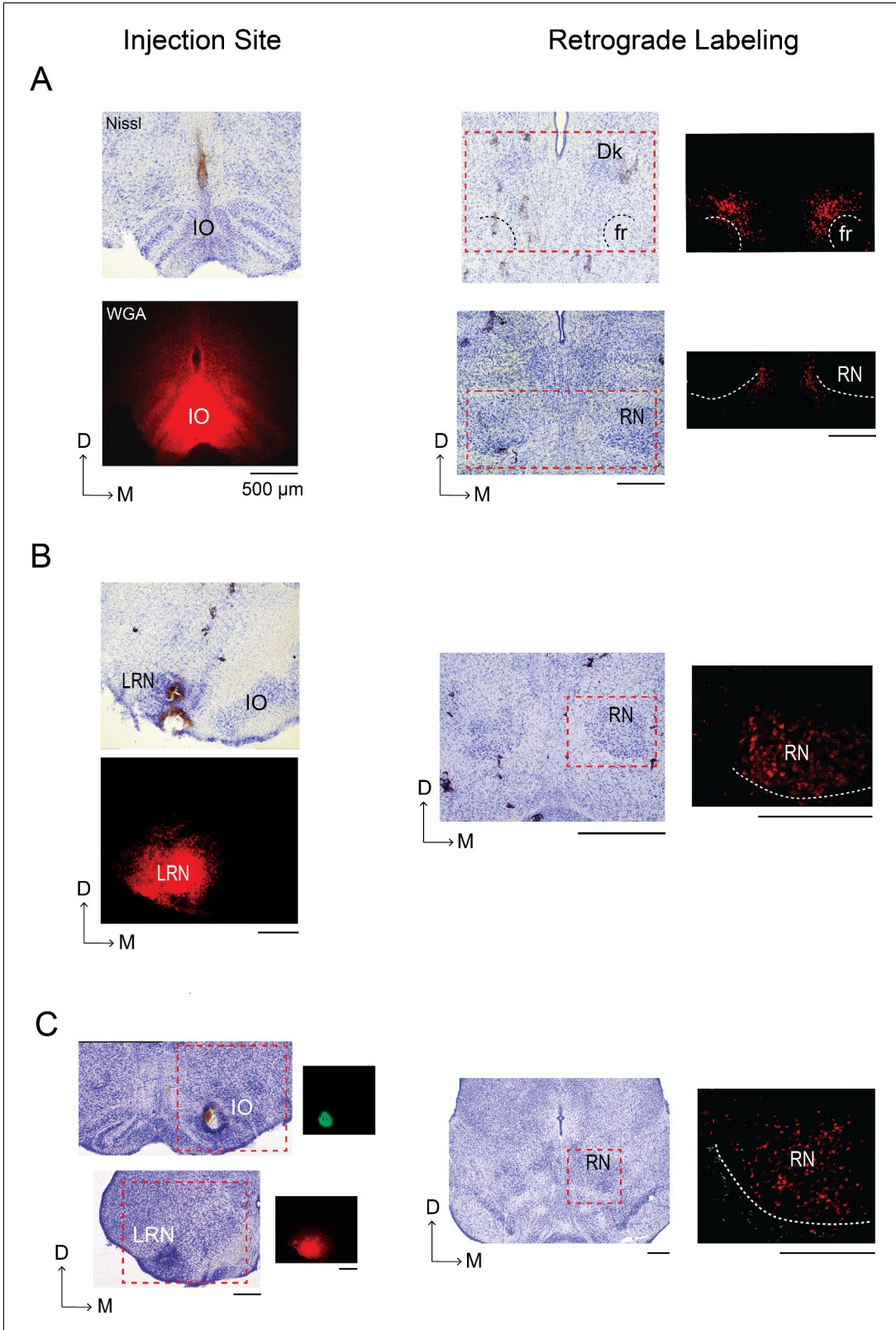

**Figure 4.** Retrograde labeling of the mesodiencephalic junction (MDJ) after infusion of WGA into the IO and LRN of P8 rats. (**A**) Left column: Coronal section depicting WGA-555 diffusion at the injection site in the IO; an adjacent Nissl-stained section is shown above. Right column: Nissl-stained coronal sections within the MDJ; retrograde labeling in the regions within the red-dashed boxes is shown at right for adjacent sections. No labeling was seen in the nucleus of Darkschewitsch (Dk) or RN, consistent with published work in adult rats (*Ruigrok et al., 2014*). White dashed lines show the

*Figure 4 continued on next page*

*Figure 4 continued*

boundaries of the fasciculus retroflexus (fr) and RN. (B) Left column: Coronal section depicting WGA-555 diffusion at the injection site in the LRN; an adjacent Nissl-stained section is shown above. Right column: Nissl-stained coronal section at the level of the RN; retrograde labeling in the region within the red-dashed box is shown at right for an adjacent section. White dashed line shows the ventromedial boundary of the RN. (C) Left column: Nissl-stained coronal sections from a single P8 rat to show the sites of WGA injection in the IO (WGA-555, red) and contralateral LRN (WGA-488, green); red-dashed boxes denote regions for the adjacent fluorescent sections shown at right. Right column: Nissl-stained coronal section at the level of the RN; retrograde labeling in the region within the red-dashed box is shown at right for an adjacent section. White dashed line shows the ventromedial boundary of the RN. D: dorsal; M: medial. All scale bars are 500 µm.

DOI: https://doi.org/10.7554/eLife.38213.010

The following figure supplement is available for figure 4:

**Figure supplement 1.** MDJ stimulation increases c-Fos expression in the IO and LRN.

DOI: https://doi.org/10.7554/eLife.38213.011

truncating IO activity and resulting in the observed sharp twitch-related peaks. To test this hypothesis, we blocked SK channels using apamin, an SK channel antagonist (*Benington et al., 1995*); apamin has been used in adult rats to block SK channels in the IO (*Lang et al., 1997*).

P8 rats were prepared for neurophysiological recording as described earlier. After the pup was cycling between sleep and wake, apamin (1 µM; dissolved in saline) or saline—mixed with 4% fluorogold to later identify the extent of diffusion—was microinjected at a volume of 100 nl into the IO. Fifteen min after the injection (the half-life of apamin is ~2 hr; *Gui et al., 2012*), the microsyringe was withdrawn and was replaced with a recording electrode (*Figure 6A*). Neural and EMG activity and sleep-wake behavior were then recorded for 30 min.

We confirmed drug or vehicle diffusion and recording sites within the IO (n = 18 units across 10 pups in the saline group; n = 21 units across 8 pups in the apamin group; 1–3 units/pup; *Figure 6B*). There was no difference in the amount of time spent in AS (p = 0.78) or in the number of twitches produced per unit time of AS between the apamin and saline groups (ps > 0.15 for nuchal, contralateral forelimb, and ipsilateral forelimb twitches; *Figure 6—figure supplement 1A,B*). As observed in the previous IO recordings, neural activity in both groups was clustered around twitches during periods of AS (*Figure 6—figure supplement 1C*). There was no significant difference in the overall firing rate during AS between groups (p = 0.59; *Figure 6—figure supplement 1D*).

Perievent histograms (10-ms bins, 1-s windows) were created for each individual unit in both groups (*Figure 6C*). As predicted, whereas twitch-triggered activity in the saline group exhibited the expected sharp peak around twitch onset, the activity in the apamin group was broader during the period after twitch onset.

The number of units exhibiting significant twitch-related activity did not differ between the two groups (n = 13/18 in saline and n = 11/21 in apamin groups; $X^2$(1, N = 39)=1.6, p = 0.2; *Figure 6D*). In contrast, the number of units exhibiting sharp peaks within ±10 ms of twitch onset was significantly lower in the apamin group than in the saline group (5/21 vs. 11/18 units, respectively; $X^2$(1, N = 39)=5.6, p = 0.02; *Figure 6D*). To illustrate the effect of apamin on twitch-related activity, we pooled the data for the significant units to create perievent histograms of IO activity. As shown in *Figure 6E*, the activity in the apamin group, unlike that in the saline group, persisted beyond 10 ms after a twitch. To quantify the difference, we calculated the area under the curve for each unit during two time windows: ±10 ms around twitch onset and 20–200 ms after twitch onset (*Figure 6F*). As expected, we found no significant difference between the two groups in the ±10-ms window (U = 56.5, Z = −0.87, p = 0.4), but did find a significant difference in the 20–200-ms window, with the apamin group being significantly larger (U = 30.5, Z = −2.2, p = 0.03). In fact, the pattern of twitch-triggered neural activity in the apamin group was similar to that recorded in the MDJ (*Figure 6G*). Based on these results, we conclude that SK channels are involved in sharpening the CD signal arriving from the MDJ.

## Discussion

Several criteria have been proposed for identifying CD signals (*Poulet and Hedwig, 2007*; *Sommer and Wurtz, 2008*). First, neurons receiving CD should increase their activity at the onset of a movement; as shown here, the activity of IO and LRN neurons increases precisely at the onset of

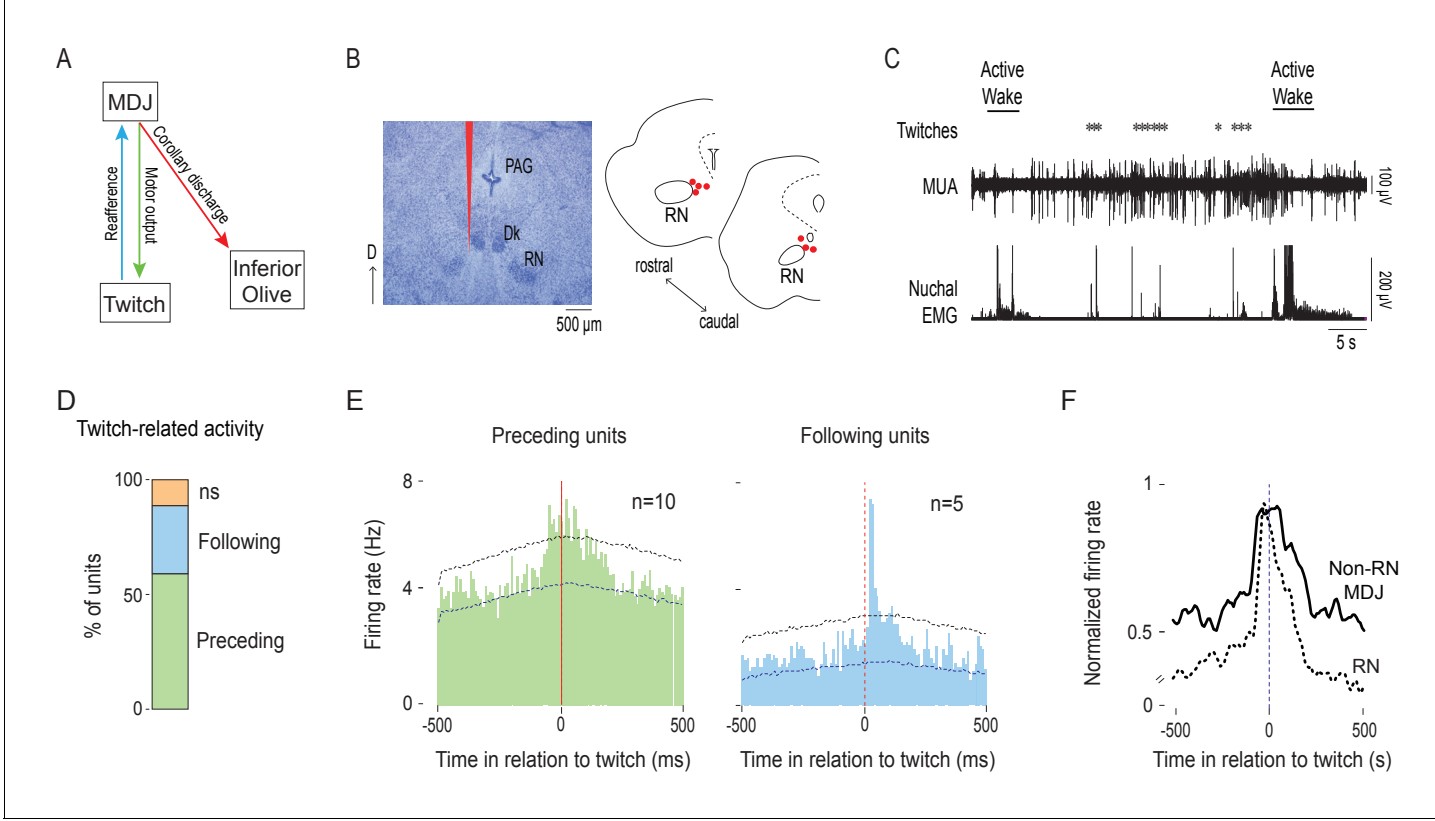

**Figure 5.** MDJ structures adjacent to the red nucleus exhibit twitch-preceding and twitch-following activity. (**A**) Diagram showing anatomical connections of the MDJ regions that lie adjacent to the red nucleus. Proposed pathways conveying motor commands (green line), reafference (blue line), and corollary discharge (red line) are shown. (**B**) Left: Representative Nissl-stained coronal brain section. Red line is the trace of a DiI-coated electrode placed within the MDJ but outside the RN. Right: Reconstruction of electrode placements (red circles) in the MDJ in two coronal sections across all pups (n = 7). D: dorsal; PAG: periaqueductal gray; Dk: nucleus of Darkschewitsch. (**C**) Representative recording of rectified nuchal EMG activity and multiunit activity (MUA) in the MDJ during spontaneous sleep-wake cycling. Asterisks denote twitches and horizontal bars denote periods of active wake movements as scored by the experimenter. (**D**) Stacked plot showing the percentage of twitch-preceding (motor; green) and twitch-following (sensory; blue) units in the MDJ. ns: not significant. (**E**) Left: Perievent histogram (10-ms bins) showing activity of twitch-preceding MDJ units in relation to twitches. Data are pooled across 10 units and triggered on 2877 twitches. Right: Perievent histogram (10-ms bins) showing activity of twitch-following MDJ units in relation to twitches. Data are pooled across 5 units and triggered on 1382 twitches. Upper and lower confidence bands (p < 0.05 for each band) are indicated by horizontal dashed lines. (**F**) Perievent histograms (10-ms bins) comparing normalized firing rate in relation to twitch onset for twitch-preceding units in the red nucleus (RN; dashed black line; data from *Del Rio-Bermudez et al., 2015*) with that of non-RN MDJ units adjacent to the red nucleus (solid black line; redrawn from E, left).

DOI: https://doi.org/10.7554/eLife.38213.012

The following source data and figure supplements are available for figure 5:

**Source data 1.** Source data for panels D-F.
DOI: https://doi.org/10.7554/eLife.38213.015
**Figure supplement 1.** Neural activity in relation to wake-movement onset.
DOI: https://doi.org/10.7554/eLife.38213.013
**Figure supplement 1—source data 1.** Source data for panels A-D.
DOI: https://doi.org/10.7554/eLife.38213.014

twitches, exhibiting a temporal profile that clearly distinguishes it from twitch-preceding activity in MDJ nuclei and twitch-following activity in the ECN. Second, a CD should originate in a structure that is demonstrably involved in the production of movement; as shown here, the twitch-related activity in the IO and LRN originates from several independent motor structures in the MDJ that are involved in the production of twitches and wake movements (*Del Rio-Bermudez et al., 2015*). Finally, areas receiving CD should themselves play no direct role in the production of movement; as precerebellar nuclei whose efferents project exclusively to the cerebellum, the IO and LRN cannot

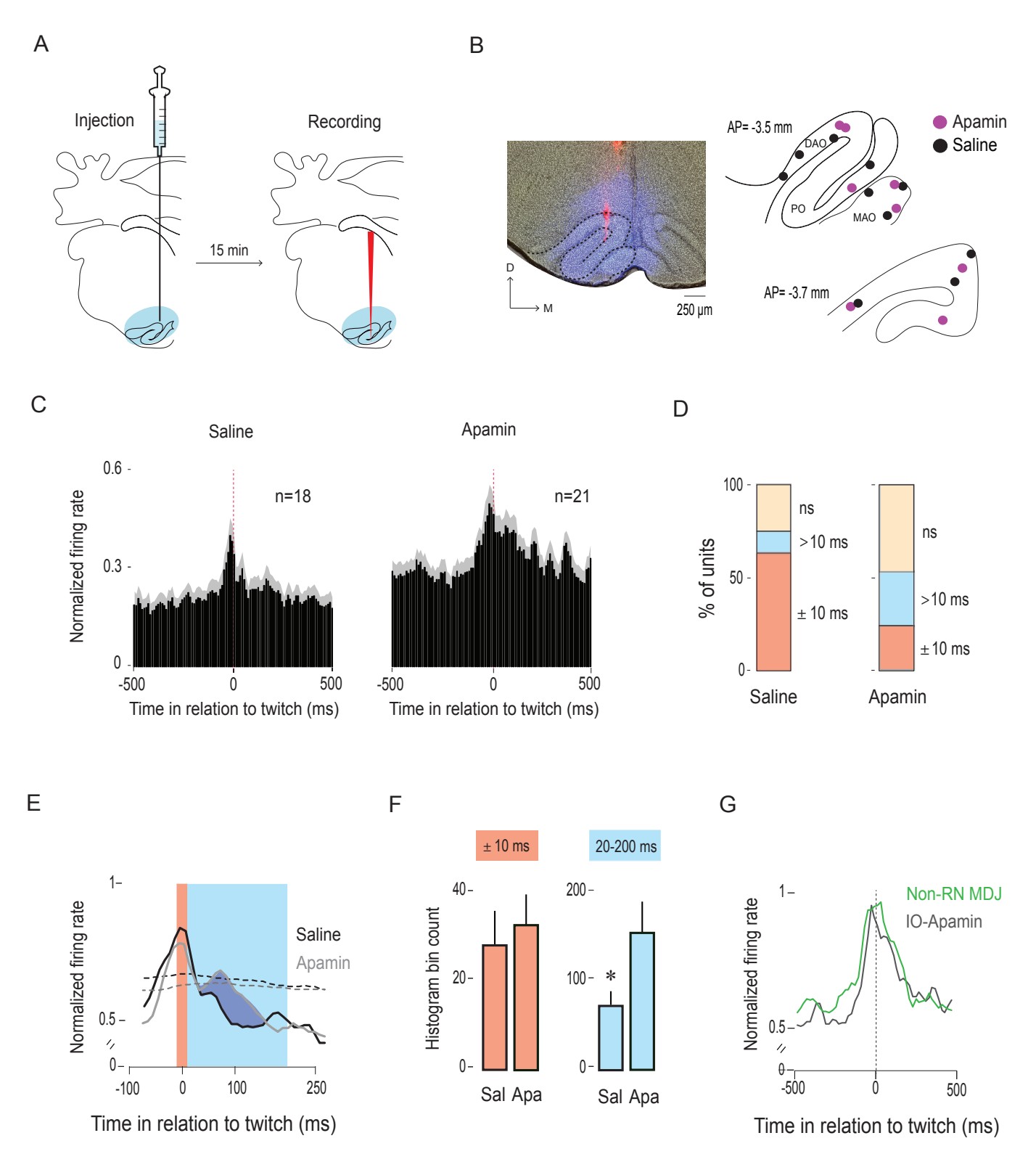

**Figure 6.** Apamin broadens the twitch-related peak in the IO. (**A**) Diagram depicting experimental design. Apamin or saline, mixed with 4% Fluorogold, was microinjected into the IO (blue shading). Fifteen minutes after the injection, the microsyringe was withdrawn and a recording electrode, coated with DiI (red vertical line), was inserted into the IO. Unit activity was recorded for 30 min. (**B**) Left: Representative coronal section showing drug diffusion in the IO (blue) and placement of DiI-coated recording electrode (red). Right: Reconstruction of electrode placements within the IO in two coronal

*Figure 6 continued on next page*

*Figure 6 continued*

sections for all pups in the saline (black dots; n = 10 pups) and apamin (purple dots; n = 8 pups) groups. DAO: dorsal accessory olive; MAO: medial accessory olive; PO: principal olive; D: dorsal; M: medial; AP: antero-posterior distance in relation to lambda. (C) Perievent histograms (10-ms bins) showing mean (+SEM; gray shading) normalized firing rates across all units triggered on twitches in the saline (n = 18) and apamin (n = 21) groups. (D) Stacked plots showing the percentage of units with significant activity within ±10 ms of twitch onset (red) and >10 ms after twitch onset (blue) in the saline and apamin groups. ns: not significant. (E) Perievent histograms (10-ms bins) showing IO unit activity in relation to twitches in the saline (black line) and apamin (gray line) groups. Data for both groups are pooled across significant units only (red and blue stacks in D; n = 13 saline units and n = 11 apamin units) and smoothed (tau = 10 ms). Red shaded area denotes ±10-ms time window around twitches. Blue shaded area denotes 20–200-ms time window following twitches. Black and gray dashed lines denote upper confidence intervals (p < 0.05) for the event correlations in the saline and apamin groups, respectively. (F) Mean histogram bin counts (area under the curve, +SEM) for firing-rate data in two time windows: the ±10-ms window around twitches (red) and the 20–200-ms window following twitches (blue) for the units in the saline (Sal; n = 12) and apamin (Apa; n = 11) groups. *p = 0.03. (G) Perievent histograms comparing normalized firing rates in relation to twitch onset for twitch-preceding units in the non-RN MDJ region (green line, same as in *Figure 5F*) with that of IO units in the apamin group (black line).

DOI: https://doi.org/10.7554/eLife.38213.016

The following source data and figure supplements are available for figure 6:

**Source data 1.** Source data for panels C-G.

DOI: https://doi.org/10.7554/eLife.38213.019

**Figure supplement 1.** Apamin does not affect sleep-wake behavior.

DOI: https://doi.org/10.7554/eLife.38213.017

**Figure supplement 1—source data 1.** Source data for panels A, B, and D.

DOI: https://doi.org/10.7554/eLife.38213.018

directly produce movement (*Gellman et al., 1985*; *Ruigrok et al., 2014*). Moreover, at P8 (and also P12) in rats, pharmacological inactivation of the deep cerebellar nuclei with muscimol exerted no discernible effects on either the rate of twitching or the duration of active sleep (*Del Rio-Bermudez et al., 2016*). Thus, the twitch-related activity in the IO and LRN satisfies the key criteria of CD. Below we discuss the implications of this finding and its significance for sensorimotor development.

## Neurophysiological identification of CD signals in behaving animals

Neural pathways conveying CD have been delineated in a diverse array of species (*Dale and Cullen, 2017*; *Davis et al., 1973*; *Fee et al., 1997*; *Schneider et al., 2014*; *Sommer and Wurtz, 2002*; *Yang et al., 2008*). Neural recordings of the CD signal itself, however, have mostly been performed in non-mammalian species, including crickets, sea slugs, crayfish, tadpoles, and electric fish (*Evans et al., 2003*; *Kirk and Wine, 1984*; *Li et al., 2004*; *Poulet and Hedwig, 2006*; *Requarth and Sawtell, 2014*). The relatively small and simple nervous systems of these species have allowed for the isolation of neurons that carry or receive CD signals and identify their relationship to behavior. In contrast, CD signals have thus far only been recorded in the mediodorsal thalamus of non-human primates during eye movements (*Sommer and Wurtz, 2004*) and in the auditory cortex of mice (*Schneider et al., 2014*).

The current findings provide the first direct neurophysiological evidence of CD in a developing mammal. Moreover, this is the first direct evidence of CD in the IO and LRN, consistent with what has been proposed for these two structures (*Alstermark and Ekerot, 2013*; *Arshavsky et al., 1978*; *De Zeeuw et al., 1998*; *Devor, 2002*). Also, with this discovery of a unique neural CD signature—comprising a short-latency onset and sharp activity peak—we have a clear template to guide future neurophysiological investigations of CD signals in other species and neural systems across the lifespan.

## A neural mechanism for sharpening the CD signal

As mentioned above, one of the signature features of the twitch-related CD signal is the sharp peak. This is surprising because, as shown here and in a previous study (*Del Rio-Bermudez et al., 2015*), twitch-related motor activity in MDJ neurons exhibits broad peaks (see *Figure 5F*). How does a broad presynaptic signal in the MDJ get converted into a sharp postsynaptic response in the IO and LRN (see *Figure 2B* and *Figure 3G*)? To answer this question, we focused on the IO because, compared with the LRN, a much higher proportion of its neurons exhibited sharp peaks.

There are a few possible candidate mechanisms. For example, in cortical pyramidal neurons, interactions between excitatory and inhibitory inputs can sharpen a neuron's activity profile (*Kremkow et al., 2010*). A similar mechanism is unlikely to operate in the IO for several reasons. First, inhibitory interneurons are sparse in that structure (<0.1%; *Nelson and Mugnaini, 1988*). Second, although the IO receives its predominant inhibitory input from the deep cerebellar nuclei (DCN; *de Zeeuw et al., 1988*), DCN activity occurs ~40 ms after a twitch (*Del Rio-Bermudez et al., 2016*). Moreover, in pilot experiments, we found that pharmacological inactivation of the DCN had no effect on IO activity at P8, consistent with a previously published report (*Nicholson and Freeman, 2003*).

Consequently, we hypothesized that inhibition in the IO is mediated by SK channels. In the IO of adult rats, these channels prevent summation of excitatory inputs (*Garden et al., 2017*). Here, using pharmacological inactivation, we demonstrate that SK channels contribute to sharpening the olivary CD signal. A similar mechanism could be functional in the LRN as SK channels are also expressed in that structure in adult rats (*Xu et al., 2013*).

## Differential actions of CD signals at precerebellar nuclei

We previously demonstrated that wake-related reafference is blocked within the ECN, likely due to modulation by a wake-related CD signal (*Tiriac and Blumberg, 2016*). This CD-mediated blockade was lifted during twitching, thereby allowing twitch-related reafference to be conveyed to downstream motor structures, including the cerebellum. In contrast, focusing here on the IO and LRN, we found that the twitch-related CD signals themselves—not reafference—are conveyed to the cerebellum (*Figure 7A*). Therefore, within this broader context, we see that CD accompanies sleep and wake behavior in infants, but its effects are not monolithic: It can modulate the action of a comparator to gate reafference (as in the ECN) or be transmitted sequentially to multiple downstream structures (as in the IO or LRN → cerebellum). Such diverse effects of CD have been described (*Crapse and Sommer, 2008*).

Although the three precerebellar nuclei—IO, LRN, and ECN—process CD and reafference differently, the common denominator of all this activity is the inundation of the developing cerebellum with twitch-related information; in contrast, activity in these nuclei is reduced during wake. As mentioned above, this reduced activity is attributable in part to wake-specific gating of reafference in the ECN (*Tiriac and Blumberg, 2016*). In addition, and as shown here, IO and LRN units were less likely to be AW-on than AS-on and exhibited weak and unreliable activity profiles around the onset of wake movements (see *Figure 1*, *Figure 3*, and *Figure 5—figure supplement 1B–D*). Altogether, the patterns of state-dependent activity in these three precerebellar nuclei are consistent with what was observed downstream in the cerebellar cortex and deep cerebellar nuclei at these ages (*Del Rio-Bermudez et al., 2016*; *Sokoloff et al., 2015a*).

## Functional implications of CD and reafference

The developing sensorimotor system receives substantial sensory input from self-generated twitches and from external stimulation arising from the mother and littermates. It has been suggested that the infant brain does not distinguish between these two sources of input and that twitch-related reafference serves merely as a 'proxy' for exafferent stimulation (*Akhmetshina et al., 2016*; *McVea et al., 2016*). This suggestion rests in part on the observation that both forms of stimulation, despite their very different origins, trigger similar patterns of cortical activity (*Akhmetshina et al., 2016*; *Tiriac et al., 2012*; *Yang et al., 2013*). With our finding that CD accompanies the production of twitches, it is now clear that there exists a mechanism with which the infant brain can distinguish self-generated from other-generated movements; the ability to make this distinction is thought to rely in part on the cerebellum (*Blakemore et al., 2000*; *Wolpert et al., 1998*).

There are a number of ways in which twitch-related CD could contribute to cerebellar development and function. For example, in the adult cerebellum, CD and reafference converge onto Purkinje cells via climbing and mossy fibers (*Blakemore et al., 2001*; *Huang et al., 2013*; *van Kan et al., 1993*; *Wolpert et al., 1998*). In this way, it is thought that the cerebellum instantiates a forward model that receives sensory predictions and computes prediction errors (by comparing CD with reafference) in order to facilitate motor learning (*Blakemore et al., 2000*; *Brooks et al., 2015*; *Requarth and Sawtell, 2014*; *Wolpert et al., 1998*).

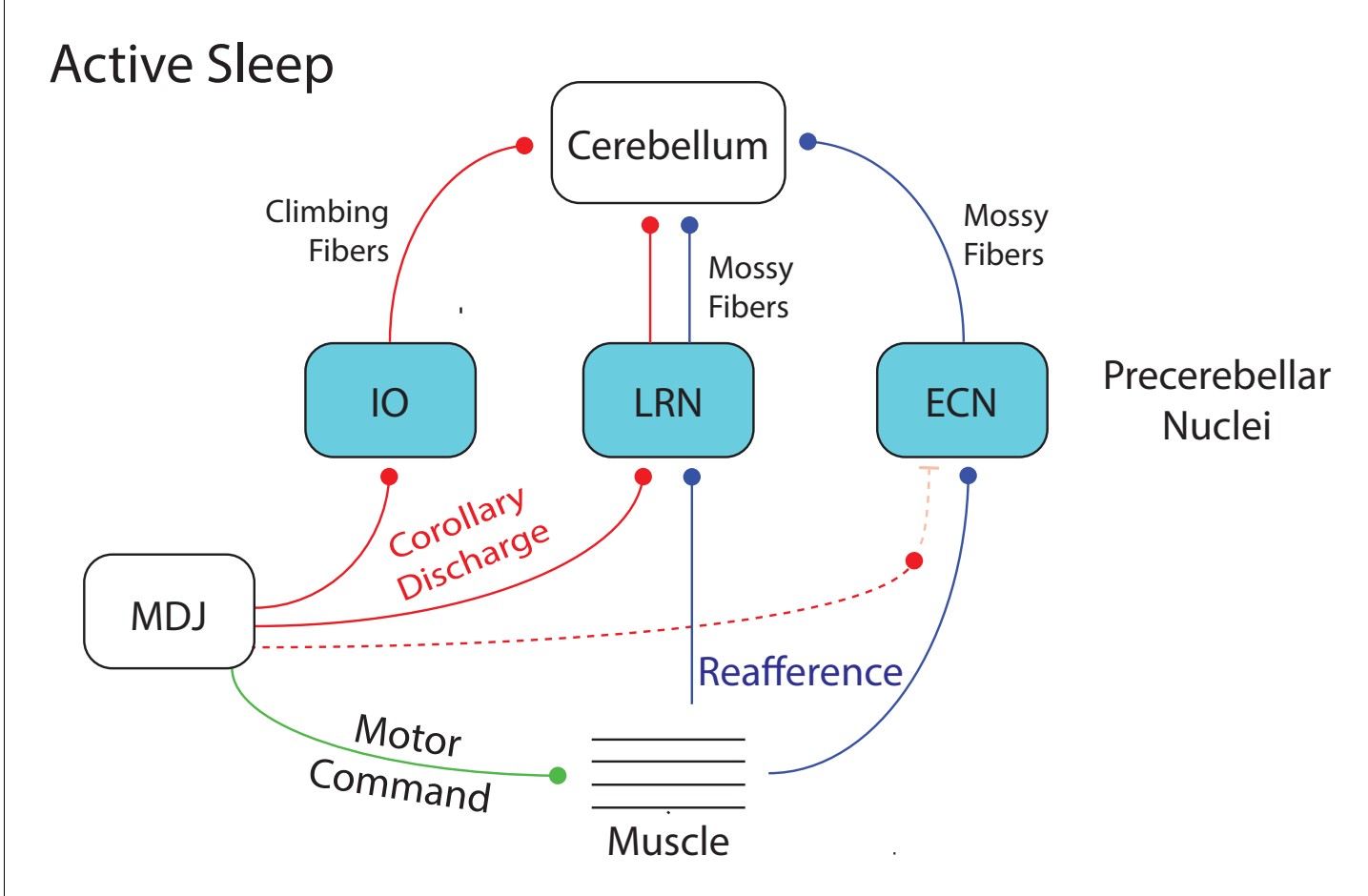

**Figure 7.** Summary diagram depicting the flow of twitch-related activity in the cerebellar system during active sleep in week-old rats. A motor command from the MDJ to muscle (green line) produces a twitch. At the same time, twitch-related CD (red lines) is conveyed from the MDJ to the cerebellum via the IO and LRN. In addition, twitch-related reafference (blue lines) is conveyed to the cerebellum via the LRN. Also, as shown previously (*Tiriac and Blumberg, 2016*), reafference to the ECN is not gated during active sleep as it is during wake, thus allowing it to flow unimpeded to the cerebellum. Dotted lines denote hypothesized connections.
DOI: https://doi.org/10.7554/eLife.38213.020

Twitches could contribute to the process by which forward models are instantiated and updated, especially in the context of a rapidly growing body. To appreciate this possibility, consider this description of cerebellar function: "After much trial and error during infancy and throughout life, the cerebellum learns to associate actual movements with intended movements. Many of our motor memories are movements that we have repeated millions or billions of times..." (p. 538, *Mason, 2011*). In that context, the millions of twitches produced in early infancy could be a critical source of repeated convergent input to the developing cerebellum. This convergence, illustrated in *Figure 7*, would provide the developing cerebellum with abundant opportunities to align prediction and feedback signals in a topographically organized fashion.

Cerebellar circuitry undergoes substantial development over the first three postnatal weeks in rats (*Altman, 1972a*; *Altman, 1972b*; *Altman, 1972c*; *Shimono et al., 1976*; *Wang and Zoghbi, 2001*). Many of these developmental processes depend heavily on neural activity, including climbing fiber synapse elimination and translocation at Purkinje cells (*Andjus et al., 2003*; *Kakizawa et al., 2000*; *Kano and Hashimoto, 2012*; *Watanabe and Kano, 2011*). With respect to synapse elimination, beginning around P8, the initial multiple innervation of Purkinje cells by climbing fibers begins to shift toward singly innervated cells in the second postnatal week as one climbing fiber is selectively strengthened over others. Importantly, spike timing-dependent plasticity (STDP) has been implicated in this process (*Kawamura et al., 2013*); STDP depends on the repetitive and sequential

firing of pre- and post-synaptic cells within a short and precise time window (*Feldman, 2012*; *Kawamura et al., 2013*; *Sgritta et al., 2017*). The present findings in precerebellar nuclei, in which twitch-related CD reliably preceded reafference by approximately 10–30 ms, are consistent with twitches playing a role in cerebellar development via STDP. In fact, recording from Purkinje cells at P8, we previously found that both complex and simple spikes were highly likely to occur within 0–50 ms after twitches (*Sokoloff et al., 2015a*).

### Implications for neurodevelopmental disorders

Disruption of cerebellar function during sensitive periods of development can have negative cascading effects on cerebello-cortical communication and ultimately on associated sensorimotor and cognitive processes, as observed in autism-spectrum disorder (*Diamond, 2000*; *Wang et al., 2014*). There are many potential causes of early cerebellar dysfunction, including prenatal and postnatal exposure to environmental stressors (*Wang et al., 2014*). One such stressor could be sleep deprivation or restriction, especially during early infancy when sleep normally predominates over wake (*Jouvet-Mounier et al., 1969*; *Roffwarg et al., 1966*). As demonstrated here and in previous studies (*Sokoloff et al., 2015a*; *Sokoloff et al., 2015b*), AS provides an important context for cerebellar activity in early development. Therefore, chronic disruptions of sleep could deprive the cerebellum and other structures of critical sensorimotor activity during sensitive periods of development.

Accumulating evidence also suggests that CD-related processing is dysfunctional in patients with schizophrenia. Specifically, failure to disambiguate 'self-generated' from 'other-generated' sensory input may underlie hallucinations and delusions of control (*Feinberg and Guazzelli, 1999*; *Ford et al., 2008*). If twitches help to instruct the developing brain to distinguish self from other, disruptions of sleep and sleep-related sensorimotor processing may have later-emerging negative consequences for the processing of CD.

### Conclusion

It has been argued that the discreteness of twitches makes them ideally suited to provide high-fidelity sensory information at ages when activity-dependent development is so important for the developing nervous system (*Blumberg et al., 2013*; *Tiriac et al., 2015*). The present results extend this idea to suggest that the convergence of twitch-related CD and reafference associated with millions of twitches over early development provides ample opportunity for assimilating growing limbs into the infant's emerging body schema (*Blumberg and Dooley, 2017*).

## Materials and methods

All experiments were carried out in accordance with the National Institutes of Health Guide for the Care and Use of Laboratory animals (NIH Publication No. 80–23). Experiments were also approved by the Institutional Animal Care and Use Committee (IACUC) of the University of Iowa.

### Subjects

Male and female Sprague-Dawley Norway rats (*Rattus norvegicus*) at postnatal day (P) 7–9 (hereafter P8; n = 68) from 60 litters were used for the study. All litters were culled to eight pups by P3. Mothers and litters were housed and raised in standard laboratory cages (48 × 20×26 cm). Food and water were available ad libitum. The animals were maintained on a 12-hr light-dark cycle with lights on at 0700 hr. Littermates were never assigned to the same experimental group.

### Surgery

A pup with a visible milk band was removed from the home cage. Under isoflurane (3–5%) anesthesia, bipolar hook electrodes (50 μm diameter, California Fine Wire, Grover Beach, CA) were inserted into the nuchal, forelimb, and hindlimb muscles for electromyography (EMG) and secured with collodion. A stainless steel ground wire was secured transdermally on the back. A custom-built head-fix device was then secured to the exposed skull with cyanoacrylate adhesive (*Blumberg et al., 2015*). The local anesthetic, bupivacaine (0.25%) was applied topically to the site of incision and some subjects were also injected subcutaneously with the analgesic agent carprofen (0.005 mg/g). The pup was lightly wrapped in gauze and allowed to recover in a humidified, temperature-controlled (35°C)

incubator for at least one hour. After recovery, the pup was briefly (<15 min) re-anesthetized with isoflurane (2–3%) and secured in a stereotaxic apparatus. A hole was drilled in the skull for insertion of the recording electrode into the inferior olive (IO; coordinates: AP = 3.4–3.6 mm caudal to lambda; ML = 0–1.2 mm), the lateral reticular nucleus (LRN; coordinates: AP = 3.5–3.7 mm caudal to lambda; ML = 1.5–1.8 mm), or midbrain nuclei near the red nucleus (RN) within the mesodiencephalic junction (MDJ; coordinates: AP = 4.7–4.9 mm caudal to bregma; ML = 0.2–0.5 mm). Two additional holes were drilled over the frontal or parietal cortices for subsequent insertion of the ground wire and a thermocouple (Omega Engineering, Stamford, CT) to measure brain temperature. In 15 pups in which no neurophysiological recordings were performed, only one additional hole was drilled for insertion of the thermocouple. After surgery, the pup was transferred to the recording chamber.

## Electrophysiological recordings

The head-fix device was secured to the stereotaxic apparatus housed within the recording chamber and the pup was positioned with its body prone on a narrow platform with limbs dangling freely on both sides (*Blumberg et al., 2015*). Care was taken to regulate air temperature and humidity such that the pup's brain temperature was maintained at 36–37°C. Adequate time (1–2 hr) was allowed for the pup to acclimate to the recording environment and testing began only when it started cycling normally between sleep and wake. Pups rarely exhibited abnormal behavior or any signs of discomfort or distress; when they did, the experiment was terminated. The bipolar EMG electrodes were connected to a differential amplifier (A-M Systems, Carlsborg, WA; amplification: 10,000x; filter setting: 300–5000 Hz). A ground wire (Ag/AgCl, 0.25 mm diameter, Medwire, Mt. Vernon, NY) was inserted into the frontal or parietal cortex contralateral to the recording site and a thermocouple was inserted into the frontal or parietal cortex ipsilateral to the recording site. Neurophysiological recordings were performed using a 16-channel silicon electrode or a four-channel linear probe (A1 × 16–10 mm-100-177; A1 × 16–8 mm-100-177; Q1 × 4–10 mm-50-177, NeuroNexus, Ann Arbor, MI), connected to a data acquisition system (Tucker-Davis Technologies, Alachua, FL) that amplified (10,000x) and filtered (500–5000 Hz) the neural signals. A digital interface and Spike2 software (Cambridge Electronic Design, Cambridge, UK) were used to acquire EMG and neurophysiological signals at 1 kHz and at least 12.5 kHz, respectively.

A micromanipulator (FHC, Bowdoinham, ME) was used to lower the electrode into the brain (DV; IO: 5.5–6.2 mm, LRN: 5–5.8 mm, midbrain structures: 4.5–4.9 mm) until action potentials were detected. Recording began at least 10 min after multiunit activity (MUA) was detected. Before insertion, the electrode was dipped in fluorescent DiI (Life Technologies, Grand Island, NY) for later identification of the recording sites. Recording of MUA and EMG activity continued for 30 min as the pup cycled freely between sleep and wake (in four pups, activity was recorded for only 15 min). The experimenter, blind to the electrophysiological record, scored the pup's sleep and wake behaviors, as described previously (*Karlsson et al., 2005*).

At the end of the recording session, the experimenter assessed evoked neural responses to exafferent stimulation of the limbs. Forelimbs and hindlimbs were gently stimulated using a paint brush. When responses to a limb stimulation were observed in at least one of the recording channels, stimulation of that limb was repeated 20–30 times at intervals of at least 5 s. Each stimulus event was marked using a key press.

## Histology

At the end of all recording sessions, pups were anesthetized with sodium pentobarbital (1.5 mg/g IP) or ketamine/xylazine (0.02 mg/g IP) and perfused transcardially with phosphate-buffered saline and 4% formalin. Brains were sectioned coronally at 80 µm using a freezing microtome (Leica Microsystems, Buffalo Grove, IL). Recording sites were determined by examining DiI tracks, before and after staining with cresyl violet, using a fluorescent microscope (Leica Microsystems, Buffalo Grove, IL).

## Retrograde tracing

Retrograde tracing was performed at P8 (n = 7) using wheat germ agglutinin (WGA) conjugated to Alexa Fluor 555 or 488 (Invitrogen Life Technologies, Carlsbad, CA). WGA-555 was injected into the

IO in three pups and into the LRN in two pups. In the remaining two pups, dual tracing was performed by injecting WGA-488 into the IO and WGA-555 into the LRN. To perform these injections, a pup was anesthetized with 2–5% isoflurane and secured in a stereotaxic apparatus. A 0.5 µl microsyringe (Hamilton, Reno, NV) was lowered stereotaxically into the IO or LRN and 0.01–0.02 µl of 2% WGA-555 or WGA-488 (dissolved in 0.9% saline) was injected over 1 min. After a 15-min post-infusion period, the microsyringe was withdrawn and the incision was closed using Vetbond (3M, Maplewood, MN). The pup was returned to its home cage and perfused 24 hr later as described above. Brains were sectioned coronally at 50 µm. Every other section was kept for Nissl staining for verification of the injection sites and areas that show retrograde labeling. Retrogradely labeled cell bodies were imaged using a fluorescent microscope (DFC300FX, Leica, Buffalo Grove, IL)

## Stimulation of MDJ structures

In urethanized (1.5 mg/g) head-fixed P8 rats (n = 8), a parylene-coated tungsten stimulating electrode (World Precision Instruments, Inc., Sarasota, FL) was lowered into the MDJ nuclei most strongly implicated by retrograde tracing. The nuclei were electrically stimulated to produce discrete movements of the forelimbs and/or hindlimbs. Trains of pulses (pulse duration: 0.2–0.4 ms; pulse frequency: 300 Hz; train width: 45 ms; *Williams et al., 2014*) were delivered every 5 s for 60 min. The current was adjusted (300–900 µA) as needed to ensure that stimulation continued to reliably produce movement. Ninety min after the last stimulation, the pup was sacrificed and the brain was prepared for c-Fos immunohistochemistry.

## Immunohistochemistry for c-Fos expression

Brains were sliced in 50 µm sections and every other section was kept for Nissl staining for verification of the stimulation sites and visualization of c-Fos expression, respectively. Primary antibody against c-Fos (anti-c-Fos rabbit polyclonal IgG; Santa Cruz Biotechnology) was diluted 1:1000 in a universal blocking serum (2% bovine serum albumin; 1% triton; 0.02% sodium azide) and applied to the sections. Sections were coverslipped and left to incubate for 48 hr at 4˚C. After incubation of the primary antibody and a series of washes in PBS, a secondary antibody (Alexa Fluor 488 donkey anti-rabbit IgG; Life Technologies, Grand Island, NY; 1:500 in PBS) was applied to the sections and incubated for 90 min at room temperature. The slides were coverslipped using Fluoro-Gel (Electron Microscopy Sciences, Hatfield, PA) and expression of c-Fos was examined using a fluorescent microscope (DFC300FX or DM6B, Leica, Buffalo Grove, IL).

## Intra-IO injection of apamin

In 18 P8 rats, pups were prepared for electrophysiological recording as described above and transferred to the recording rig. Once a pup started cycling between sleep and wake, a 0.5 µl microsyringe was lowered stereotaxically into the IO and 100 nl of apamin (Abcam, Cambridge, MA; 1 µM, dissolved in 0.9% saline, n = 8) or saline (n = 10) was injected over 1 min. During preparation of the drug or vehicle, fluorogold (4%, Fluorochrome, Denver, CO) was added to the solutions for subsequent assessment of the extent of drug diffusion. After a 15-min period to allow for diffusion, the microsyringe was withdrawn and a recording electrode was lowered in its place into the IO and activity was recorded for 30 min. At the end of the experiment, the pup was sacrificed and its brain was prepared for histology as described above.

## Data analysis

### Spike sorting

As described previously (*Mukherjee et al., 2017*; *Sokoloff et al., 2015a*), action potentials (signal-to-noise≥2:1) were sorted from MUA records using template matching and principal component analysis in Spike2 (Cambridge Electronic Design). Waveforms exceeding 3.5 SD from the mean of a given template were excluded from analysis.

### Identification of behavioral states

EMG activity and behavioral scoring were used to identify behavioral state (*Blumberg et al., 2015*). To establish an EMG threshold for distinguishing sleep from wake, EMG signals were rectified and smoothed (tau = 0.001 s). The mean amplitude of high muscle tone and atonia were calculated from

five representative 1-s segments and the midpoint between the two was used to establish the threshold for defining periods of wake (defined as muscle tone being above the threshold for at least 1 s) and sleep (defined as muscle tone being below the threshold for at least 1 s). Active wake (AW) was identified by high-amplitude limb movements (e.g. stepping, stretching) against a background of high muscle tone and was confirmed using behavioral scoring. The onset of a wake movement was defined on the basis of EMG amplitude surpassing the established threshold. Active sleep (AS) was characterized by the presence of myoclonic twitches of the limbs against a background of muscle atonia. Twitches were identified as sharp EMG events that exceeded by ≥3 x the mean EMG baseline during atonia; twitches were also confirmed by behavioral scoring (*Seelke and Blumberg, 2010*). Additionally, behavioral quiescence (BQ) was characterized as periods of low muscle tone interposed between AW and AS.

## State-dependent neural activity

For each unit, average firing rate across all behavioral states was determined. Bouts of AS, AW, and BQ were excluded when firing rates exceeded 3 SD of the firing rate for that behavioral state; this happened rarely (0–2 per unit). Next, pairwise comparison of firing rates across states was performed using the Wilcoxon matched-pairs signed-ranks test (SPSS; IBM, Armonk, NY). Units were categorized as AS-on (AS >AW ≥ BQ), AW-on (AW >AS ≥ BQ), AS+AW-on (AS = AW > BQ) or state-independent (AS = AW = BQ). Firing rates of all AS-on and AS+AW-on units across behavioral states were further compared using the Wilcoxon matched-pairs signed-ranks test.

## Twitch-triggered neural activity

To determine the relationship between unit activity and twitching, we triggered unit activity on twitch onsets and generated perievent histograms over a 1-s window using 5- or 10-ms bins. We performed these analyses on each individual unit using twitches from nuchal, forelimb, and hindlimb muscles. We tested statistical significance by jittering twitch events 1000 times over a 500-ms window using PatternJitter (*Amarasingham et al., 2012*; *Harrison and Geman, 2009*), implemented in MATLAB (MathWorks, Natick, MA), which generates upper and lower confidence bands (p < 0.05 or 0.01 for each band) using a method that corrects for multiple comparisons. For each unit, after histograms were separately constructed for nuchal, forelimb, or hindlimb twitches, we identified activity that was significant in response to a twitch. When more than one muscle yielded a significant change in neural activity, we further analyzed the data only for the muscle that showed the strongest relationship (determined by the highest firing rate) between twitches and unit activity. We then pooled these data to create perievent histograms composed of significant units and performed final jitter analyses on the pooled data.

## Wake-triggered neural activity

To determine the relationship between neural activity and wake movements, we triggered unit activity on wake-movement onset and, as described above, used jitter analyses to determine which units were individually significant. When appropriate, we pooled these data to create perievent histograms composed of significant units and performed final jitter analyses on the pooled data.

## Evoked response to exafferent stimulation

We identified MUA in which evoked responses were observed and then sorted the units. Those units were then pooled and triggered on stimulus onset (determined using EMG artifact) to create perievent histograms. The jitter analysis was performed on the individual unit data, as described above.

## Intra-olivary injection of apamin

First, we identified if apamin affected sleep-wake behavior. We assessed the amount of time spent in AS and the number of twitches per min of AS in each pup. Differences across groups were tested using the Mann-Whitney *U* test. Next, we determined if apamin altered the overall firing rate. We calculated the firing rate of each unit during AS and compared that across groups using the Mann-Whitney *U* test. One value exceeding 3 SD was excluded as an outlier.

We then assessed whether apamin altered the shape of twitch-triggered perievent histogram. First, we created perievent histograms (10-ms bins, 1-s window) for each unit as described above.

For each unit, firing rate was normalized to the peak firing rate and the average normalized firing rate across all units in each group was calculated. Perievent histograms were then created with the average (+SEM) normalized firing rates triggered on twitches for each group. Next, we assessed how apamin altered the pattern of twitch-triggered activity of individual units. To do that, we identified significant units by performing jitter analysis on individual units as described above. We counted the percentage of units that showed a precise peak within ±10 ms of twitch onset and compared that across groups using a Chi-squared test. Finally, we pooled significant units in each group and pooled them to create perievent histograms consisting of significant units only. To assess the difference in the shape of perievent histograms, we calculated the area under the curve by adding the histogram counts within a particular time window and compared that across groups using the Mann-Whitney $U$ test. One value exceeding 3 SD was excluded as an outlier.

Unless otherwise stated, alpha was set at 0.05.

## Data availability

Source data files have been provided for *Figures 1*, *2*, *3*, *5* and *6*.

## Acknowledgements

The authors thank Alex Tiriac and Jimmy Dooley for helpful comments and technical assistance. Research was supported bygrants from the National Institute of Health (R37 HD-081168 and R01 HD-063071) to MSB. The authors declare no competing financial interests.

## Additional information

### Funding

| Funder | Grant reference number | Author |
| --- | --- | --- |
| Eunice Kennedy Shriver National Institute of Child Health and Human Development | R37-HD081168 | Mark S Blumberg |
| Eunice Kennedy Shriver National Institute of Child Health and Human Development | R01-HD063071 | Mark S Blumberg |

The funders had no role in study design, data collection and interpretation, or the decision to submit the work for publication.

### Author contributions

Didhiti Mukherjee, Conceptualization, Data curation, Formal analysis, Validation, Investigation, Visualization, Methodology, Writing—original draft, Project administration, Writing—review and editing; Greta Sokoloff, Conceptualization, Data curation, Supervision, Funding acquisition, Visualization, Methodology, Project administration, Writing—review and editing; Mark S Blumberg, Conceptualization, Supervision, Funding acquisition, Methodology, Project administration, Writing—review and editing

### Author ORCIDs

Didhiti Mukherjee https://orcid.org/0000-0003-3660-2831
Greta Sokoloff https://orcid.org/0000-0003-3651-1980
Mark S Blumberg https://orcid.org/0000-0001-6969-2955

### Ethics

Animal experimentation: All experiments were carried out in accordance with the National Institutes of Health Guide for the Care and Use of Laboratory Animals (NIH Publication No. 80-23) and were approved by the Institutional Animal Care and Use Committee of the University of Iowa (protocol numbers 1403038 and 7011955).

Decision letter and Author response
Decision letter https://doi.org/10.7554/eLife.38213.025
Author response https://doi.org/10.7554/eLife.38213.026

# Additional files

## Supplementary files
• Transparent reporting form
DOI: https://doi.org/10.7554/eLife.38213.021

## Data availability

Our neurophysiological data is available on Dryad under DOI10.5061/dryad.rf524vv.

The following dataset was generated:

| Author(s) | Year | Dataset title | Dataset URL | Database and Identifier |
|---|---|---|---|---|
| Didhiti Mukherjee, Greta Sokoloff, Mark S Blumberg | 2018 | Data from: Corollary discharge in precerebellar nuclei of sleeping infant rats | http://doi.org/10.5061/dryad.rf524vv | Dryad Digital Repository, 10.5061/dryad.rf524vv |

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
