## [Decision Letter]

[Editors’ note: the authors were asked to provide a plan for revisions before the editors issued a final decision. What follows is the editors’ letter requesting such plan.]

Thank you for sending your article entitled "It's not you, it's me: Corollary discharge in precerebellar nuclei of sleeping infant rats" for peer review at *eLife*. Your article is being evaluated by Sabine Kastner as the Senior Editor, a Reviewing Editor, and three reviewers.

Given the list of essential revisions, which may potentially include new experiments, the editors and reviewers invite you to respond within the next two weeks with an action plan and timetable for the completion of the additional work. We plan to share your responses with the reviewers and then issue a binding recommendation.

The reviewers were generally enthusiastic about the importance of this work. However, all reviewers felt that one finding was not supported by replications in a sufficient number of mice (i.e., Figure 4C), although reviewers all agreed that the authors could choose to deal with this issue in a few different ways (see reviewer #1, Essential revision 1). Reviewers also agreed that more thorough analyses of neuronal recordings, particularly during awake movements, would strengthen the paper. Moreover, it was agreed that the authors should include confidence intervals, or error bars, on figures where appropriate (e.g.,, Figure 2D, Figure 5F, Figure 6E, Figure 6G).

However, during the discussion, reviewers were unable to come to an agreement on another major point. One reviewer strongly felt that a microstimulation experiment was essential to prove that the observed activity is a corollary discharge and not associated with the generation of twitch movement (reviewer #3, Essential revision 3). The second reviewer strongly disagreed (i.e., thought that this experiment was not essential to the paper's main conclusions), and the first reviewer did not have a strong opinion either way. So, in the end, the Reviewing Editor and the reviewers decided to let the authors choose how to address this point and to explain their decision in their action plan. The reviews are included in their entirety below.

*Reviewer #1:*

Sleep during development is important for shaping sensorimotor development. This study investigates neural circuits underlying sensorimotor development. The study investigates the extent to which twitches occurring in infant rats are just random muscle movements without any further relevance for the developing motor system or instead internally generated movement which share the same neuronal substrates as movements during wakefulness, including corollary discharge and efference copies.

Essential revisions:

1) Figure 4C used only n=2. This is a low number of rats that does not support a concrete conclusion. When presenting this result, the authors should either include more rats to increase their n, remove this finding, or write a caveat in the Results section (since these experiments in which tracers were injected into two brain areas in the same rat simply bolster the results that were already shown with single tracer injections in each of the two brain areas in two different, larger groups of rats).

2) Figure 1B, Figure 3B, Figure 5B: It is unclear whether these histology pictures are really from rats in which units were recorded. To better help the readers to understand where the recording was performed, it would be informative to add the actual histology picture showing the position of the electrode.

3) Figure 5—figure supplement 1B: Why is the wake movement related perievent histogram so flat compared to the very sharp twitch related histogram in Figure 2? Do the IO neurons only fire in response to specific movements such that in average the peak is blurred out? If so, what subclass of movements is accompanied by IO spiking? For example, do only forelimb movements show CD? To convincingly show that the IO is part of the "normal" pathway how internally generated movements are triggered, I would assume that these neurons should show similar precise and discharge during wake movements.

To stress the point that the IO is a (potentially) unique brain area in the mammalian brain showing CD the authors should invest a bit more into explaining what happens during wakefulness. Which types of movements are accompanied by CD, how reliable is the CD, etc.? Perhaps some example raw traces could be shown. Overall such data would further stress the significance of CD during active sleep for development. (If the IO only shows CD during active sleep, but not during wake, it's probably not that important for development). Similarly, it would be interesting for the authors to also perform more analyses for the neuronal activity in LRN and MDJ during wakefulness.

4) "Moreover, of the 7 IO units that exhibited twitch-triggered responses at latencies consistent with reafferent processing (i.e.,, > 10 ms; Figure 2—figure supplement 1E), 3 responded to exafferent stimulation."

Figure 2—figure supplement 1E does not show the data for exafferent stimulation. It shows the twitch-related perievent histogram. Could the authors please show the perievent histogram in relation to (external) movements of the forelimb to clearly stress the crucial point that "red" IO units only respond to twitches but not to externally triggered (exafferent) movement?

*Reviewer #2:*

Summary:

The Blumberg lab's previous work showed that while awake movements elicit a corollary discharge in the external cuneate nucleus, sleep twitches do not. The question addressed in the current study is whether the lack of a corollary discharge following sleep twitches is a general, nervous system wide phenomenon or limited to the external cuneate nucleus. The latter was suggested by an experiment that showed that sleep twitches elicited complex and simple spikes in the cerebellum. This brings us to the current study: Investigating whether or not the sources for climbing fiber and mossy fiber pathways (inferior olive and lateral reticular nucleus, respectively) show evidence of sleep twitch-related corollary discharge.

Using a combination of approaches – electrophysiology, microstimulation, immediate-early gene expression, tracers, and pharmacology – the authors provide converging (and convincing) evidence that the both the inferior olive and the lateral reticular nucleus receive a corollary discharge following sleep-twitches (with the lateral reticular nucleus also showing longer-latency reafferent responses in some neurons).

This work is another important piece of the puzzle of sleep twitches and their role in nervous system development. I don't have any issues with the experiments or their interpretations. No single approach is entirely convincing but when taken together, it is difficult to reach any other conclusion than the existence of sleep twitch corollary discharge in the inferior olive and lateral reticular nucleus.

Essential revisions:

1) The number of neurons for some conditions seems to be a bit on the low side (e.g., 16 neurons for the LRN experiments). The authors should explain why this is so (e.g., difficult to record from certain structures?) and provide effect sizes.

2) It would be nice to have the cresyl violet sections for the tracer results shown in Figure 4.

3) The Introduction does not do justice to the importance of the study. In its current form it quickly jumps into the Blumberg's lab previous studies as a source of motivation. It seems like one could broaden the audience by introducing first the importance of twitches during development to the self-organizing nervous system, then the putative importance of corollary discharges in this process.

4) The reader might better follow the logic of the results if the paragraph that starts the Discussion section (about how to identify a corollary discharge signal) was moved to the beginning of the Results section.

5) The Abstract does not do justice to the many approaches taken by the authors to support their discovery. As it stands, it seems like they only did some recordings and the apamin experiment.

*Reviewer #3:*

This study builds on prior work from the same laboratory on the role of active sleep-associated brain activity in generation of twitches during nervous system development. The authors here provide a descriptive overview of twitch-associated discharge in inferior olive and lateral reticular nuclei of the cerebellum. They demonstrate that the firing of large subpopulations of neurons in the IO and LRN is higher during active sleep than during waking and is locked to the execution of twitches during sleep. Some additional characterization of the described results is essential for data interpretation. Specific suggestions are outlined below:

Essential revisions:

1) It would be useful to better characterize certain parameters described in the results. For example, it would be nice to see the ratio of AS firing to awake firing for individual units classified as AS-on, AW-on and AS+AW-on. It would also be helpful to show a quantification of the ratio of twitch-associated to non-twitch associated firing for each cell.

2) What about the AW-on cells in IO and LRN? How does their activity correlate with movements? Because the major novel finding is the discrimination of AS twitch associated activity (as opposed to waking movement activity), it would be nice to see a comparison of these two phenomena.

3) The authors claim, based on prior work, that the activity of IO and LRN is not responsible for generating or altering movements. It would be very useful to demonstrate this in the system under study, across AS and AW. The authors do stimulation studies as described above, but effects of stimulation of IO and LRN in different states, to fully characterize effects (or lack thereof) on movement would make the firing correlations far more meaningful.

4) On a related note, the authors use apamin to test how correlated firing is driven with respect to (twitch) movements. A quantification of the relationship of this firing effect to movement (if any) would be very helpful (i.e., an analysis of the temporal relationship of firing, relative to both twitches *and* movements during waking, under control and apamin-treated conditions). This would further bolster the argument that the firing of these neurons is a pure CD and not a driver of movement.

[Editors' note: further revisions were requested prior to acceptance, as described below.]

Thank you for resubmitting your work entitled "Corollary discharge in precerebellar nuclei of sleeping infant rats" for further consideration at *eLife*. Your revised article has been favorably evaluated by Richard Ivry (Senior Editor), Laura Colgin (Reviewing Editor), and two reviewers.

The manuscript has been improved but there are some remaining issues that need to be addressed before acceptance, as outlined below:

Essential revisions:

Figure 5—figure supplement 1 and Figure 7: In the response, the authors claim that they performed more thorough analyses of wake-related activity. However, the authors mentioned that they could not detect wake related activity because wake movements are rare at this age. Therefore, it is still unclear what exactly happens during wake or whether the findings for twitches translate to active wakefulness.

And consequently, in Figure 7, the line from MDJ to LRN during wake movements is not supported by any of authors' findings. Also, in the Figure 7 legend, the authors mention: activity is less pronounced or actively suppressed during wake movements in these three precerebellar nuclei, thereby reducing input to the cerebellum. Authors didn't provide data to support this claim.

---

## [Author Response]

[Editors' note: the authors’ plan for revisions was approved and the authors made a formal revised submission.]

The reviewers were generally enthusiastic about the importance of this work. However, all reviewers felt that one finding was not supported by replications in a sufficient number of mice (i.e., Figure 4C), although reviewers all agreed that the authors could choose to deal with this issue in a few different ways (see reviewer #1, Essential revision 1). Reviewers also agreed that more thorough analyses of neuronal recordings, particularly during awake movements, would strengthen the paper. Moreover, it was agreed that the authors should include confidence intervals, or error bars, on figures where appropriate (e.g., Figure 2D, Figure 5F, Figure 6E, Figure 6G).However, during the discussion, reviewers were unable to come to an agreement on another major point. One reviewer strongly felt that a microstimulation experiment was essential to prove that the observed activity is a corollary discharge and not associated with the generation of twitch movement (reviewer #3, Essential revision 3). The second reviewer strongly disagreed (i.e., thought that this experiment was not essential to the paper's main conclusions), and the first reviewer did not have a strong opinion either way. So, in the end, the Reviewing Editor and the reviewers decided to let the authors choose how to address this point and to explain their decision in their action plan. The reviews are included in their entirety below.Reviewer #1:Essential revisions:1) Figure 4C used only n=2. This is a low number of rats that does not support a concrete conclusion. When presenting this result, the authors should either include more rats to increase their n, remove this finding, or write a caveat in the Results section (since these experiments in which tracers were injected into two brain areas in the same rat simply bolster the results that were already shown with single tracer injections in each of the two brain areas in two different, larger groups of rats).

We understand the reviewer’s concern and have added language to the text to clarify the number of subjects used in this tracing study in the subsection “Non-overlapping regions in the mesodiencephalic junction (MDJ) project to the IO and LRN”.

2) Figure 1B, Figure 3B, Figure 5B: It is unclear whether these histology pictures are really from rats in which units were recorded. To better help the readers to understand where the recording was performed, it would be informative to add the actual histology picture showing the position of the electrode.

As recommended, we have added Nissl-stained images in Figure 1B, Figure 3B, and Figure 5B to show the position of the recording electrodes.

3) Figure 5—figure supplement 1B: Why is the wake movement related perievent histogram so flat compared to the very sharp twitch related histogram in Figure 2? Do the IO neurons only fire in response to specific movements such that in average the peak is blurred out? If so, what subclass of movements is accompanied by IO spiking? For example, do only forelimb movements show CD? To convincingly show that the IO is part of the "normal" pathway how internally generated movements are triggered, I would assume that these neurons should show similar precise and discharge during wake movements.To stress the point that the IO is a (potentially) unique brain area in the mammalian brain showing CD the authors should invest a bit more into explaining what happens during wakefulness. Which types of movements are accompanied by CD, how reliable is the CD, etc.? Perhaps some example raw traces could be shown. Overall such data would further stress the significance of CD during active sleep for development. (If the IO only shows CD during active sleep, but not during wake, it's probably not that important for development). Similarly, it would be interesting for the authors to also perform more analyses for the neuronal activity in LRN and MDJ during wakefulness.

As suggested by the reviewer, we have now performed more thorough analyses of wake-related activity and added that information to the manuscript (see Figure 5—figure supplement 1). In general, wake-related activity in the IO and LRN is weaker and less reliable than for twitch-related activity (see subsection “MDJ neurons adjacent to the RN are active before and after the production of twitches”).

4) "Moreover, of the 7 IO units that exhibited twitch-triggered responses at latencies consistent with reafferent processing (i.e.,, > 10 ms; Figure 2—figure supplement 1E), 3 responded to exafferent stimulation."Figure 2—figure supplement 1E does not show the data for exafferent stimulation. It shows the twitch-related perievent histogram. Could the authors please show the perievent histogram in relation to (external) movements of the forelimb to clearly stress the crucial point that "red" IO units only respond to twitches but not to externally triggered (exafferent) movement?

As suggested, we have added a perievent histogram to show IO activity in relation to exafferent stimulation for the CD (red) units; see Figure 2—figure supplement 1E. As expected, we did not see IO responses to exafferent stimulation.

Reviewer #2:Essential revisions:1) The number of neurons for some conditions seems to be a bit on the low side (e.g., 16 neurons for the LRN experiments). The authors should explain why this is so (e.g., difficult to record from certain structures?) and provide effect sizes.

As the reviewer correctly suspects, it is very difficult to identify, hold, and record for 30 minutes from ventral brainstem structures like the IO and LRN, especially in unanesthetized pups. Typically, in the past, we have reported ~15-20 units from 6-8 pups for each brain structure. In the current study, we have used more than 6 pups in each electrophysiological experiment and in each group.

2) It would be nice to have the cresyl violet sections for the tracer results shown in Figure 4.

As suggested, we have added cresyl violet sections to Figure 4 and to Figure 4—figure supplement 1.

3) The Introduction does not do justice to the importance of the study. In its current form it quickly jumps into the Blumberg's lab previous studies as a source of motivation. It seems like one could broaden the audience by introducing first the importance of twitches during development to the self-organizing nervous system, then the putative importance of corollary discharges in this process.

We agree; the Introduction has been edited so that the potential significance of our findings is made more clear and explicit.

4) The reader might better follow the logic of the results if the paragraph that starts the Discussion section (about how to identify a corollary discharge signal) was moved to the beginning of the Results section.

We agree that a more explicit discussion of the logic of the study should come earlier in the paper. Accordingly, we revised key paragraphs at the end of the Introduction to describe the experiments within the context of the criteria proposed for identifying CD.

5) The Abstract does not do justice to the many approaches taken by the authors to support their discovery. As it stands, it seems like they only did some recordings and the apamin experiment.

We completely agree, and the Abstract now provides substantially more detail.

Reviewer #3:Essential revisions:1) It would be useful to better characterize certain parameters described in the results. For example, it would be nice to see the ratio of AS firing to awake firing for individual units classified as AS-on, AW-on and AS+AW-on. It would also be helpful to show a quantification of the ratio of twitch-associated to non-twitch associated firing for each cell.

As suggested, we have now quantified state-dependent firing of AS+AW-on and AW-on units and added that information to the manuscript and show it in Figures 1F and 3E (subsections “IO activity predominates during active sleep” and “LRN neurons exhibit two kinds of twitch-related activity”).

As mentioned above in response to reviewer 1, EMGs are limited in their ability to detect all twitches within a muscle (and limb). Therefore, we must be cautious when interpreting results using this method. We are currently working on better methods for assessing the reliability of twitches to activate neural activity.

2) What about the AW-on cells in IO and LRN? How does their activity correlate with movements? Because the major novel finding is the discrimination of AS twitch associated activity (as opposed to waking movement activity), it would be nice to see a comparison of these two phenomena.

We have now quantified state-dependent firing of AW-on cells in the IO and LRN and added that information to the manuscript. AW-on cells, however, were extremely rare in both the IO (2/35 units) and LRN (2/27 units) at this age. In contrast with AS-on units, in which activity often decreased after wake onset, AW-on cells exhibited persistent activity throughout wake bouts. Importantly, the activity during wake was significantly higher than that during the other two states (subsections “IO activity predominates during active sleep” and “LRN neurons exhibit two kinds of twitch-related activity”.

3) The authors claim, based on prior work, that the activity of IO and LRN is not responsible for generating or altering movements. It would be very useful to demonstrate this in the system under study, across AS and AW. The authors do stimulation studies as described above, but effects of stimulation of IO and LRN in different states, to fully characterize effects (or lack thereof) on movement would make the firing correlations far more meaningful.

We felt we had addressed this issue by noting that the IO and LRN project exclusively to the cerebellum and citing a paper showing that stimulation of the IO in adults does not elicit movements. It seems, however, that this reviewer was not convinced and remains concerned that the IO and LRN could influence movements via their projections to the cerebellum.Accordingly, we have revised the text to discuss this issue more fully, including adding a citation to a paper from our lab in which the deep cerebellar nuclei (the sole outputs of the cerebellum) were inactivated in P8 rats without having an effect on twitching or active sleep (Discussion section).

*4) On a related note, the authors use apamin to test how correlated firing is driven with respect to (twitch) movements. A quantification of the relationship of this firing effect to movement (if any) would be very helpful (i.e., an analysis of the temporal relationship of firing, relative to both twitches* and *movements during waking, under control and apamin-treated conditions). This would further bolster the argument that the firing of these neurons is a pure CD and not a driver of movement.*

The apamin experiment was meant to test whether the sharp twitch-related peaks were driven by opening of SK channels. The activity of the IO neurons was clustered around twitches during periods of active sleep (see Figure 6—figure supplement 1C) and, was substantially decreased during wakefulness in both saline- and apamin-treated groups. Accordingly, we restricted our analyses to twitch-related activity.

[Editors' note: further revisions were requested prior to acceptance, as described below.]

The manuscript has been improved but there are some remaining issues that need to be addressed before acceptance, as outlined below:Essential revisions:Figure 5—figure supplement 1 and Figure 7: In the response, the authors claim that they performed more thorough analyses of wake-related activity. However, the authors mentioned that they could not detect wake related activity because wake movements are rare at this age. Therefore, it is still unclear what exactly happens during wake or whether the findings for twitches translate to active wakefulness.

And consequently, in Figure 7, the line from MDJ to LRN during wake movements is not supported by any of authors' findings. Also, in the Figure 7 legend, the authors mention: activity is less pronounced or actively suppressed during wake movements in these three precerebellar nuclei, thereby reducing input to the cerebellum. Authors didn't provide data to support this claim.

We were insufficiently clear in presenting the results for wake movements. It is certainly true that the lower frequency of wake movements creates some difficulties for analysis, but the lower frequency of wake movements also highlights the relative significance of twitching. This significance is supported further by the fact that wake-related activity in the precerebellar nuclei is either actively suppressed (see Tiriac and Blumberg,2016) or relatively weak and unreliable (as shown here).

In discussing wake-related activity in the paper, we did not sufficiently emphasize how our results are consistent with what we have reported previously downstream in the cerebellum: that neural activity is high during active sleep and low during wake. Accordingly, we have edited the text to highlight these facts and communicate the results using more consistent language. See subsection “MDJ neurons adjacent to the RN are active before and after the production of twitches”, subsection “Differential actions of CD signals at precerebellar nuclei”, and Figure 5—figure supplement 1 legend.

We have also come to believe that the inclusion of Figure 7B detracted from the main focus of the paper, which is communicated most clearly in Figure 7A. Accordingly, we have deleted Figure 7B. The text still addresses wake-related activity, but in a more clear and consistent manner.